# IL-36 signaling in keratinocytes controls early IL-23 production in psoriasis-like dermatitis

Jérémie D Goldstein[1], Esen Y Bassoy[1], Assunta Caruso[1], Jennifer Palomo[1], Emiliana Rodriguez[1], Sylvain Lemeille[1], Cem Gabay[1,2]

IL-36R signaling plays an important role in the pathogenesis of psoriasis. We ought to assess the specific function of IL-36R in keratinocytes for the pathology of Aldara-induced psoriasis-like dermatitis. $Il36r^{\Delta K}$ mice presenting deletion of IL-36R in keratinocytes were similarly resistant to Aldara-induced ear inflammation as $Il36r^{-/-}$ mice, but acanthosis was only prevented in $Il36r^{-/-}$ mice. FACS analysis revealed that IL-36R signaling in keratinocytes is mandatory for early neutrophil infiltration in Aldara-treated ears. RNASeq and qRT-PCR experiments demonstrated the crucial role of IL-36R signaling in keratinocytes for induction of IL-23, IL-17, and IL-22 at early time points. Taken together, our results demonstrate that IL-36R signaling in keratinocytes plays a major role in the induction of Aldara-induced psoriasis-like dermatitis by triggering early production of IL-23/IL-17/IL-22 cytokines and neutrophil infiltration.

## Introduction

Psoriasis is a chronic inflammatory disease which affects around 1–2% of the world's population. It is characterized by epidermal hyperplasia (acanthosis), formation of erythematous plaques, dermo-epidermal inflammation, and angiogenesis. These manifestations are caused by infiltration of neutrophils and inflammatory mononuclear cells in the dermis and epidermis (Lowes et al, 2014; Boehncke & Schon, 2015). The IL-23/IL-17/IL-22 axis is fundamental for the development of the disease (Lowes et al, 2014), as demonstrated by the efficacy of therapies targeting IL-17A (Griffiths et al, 2015; Lebwohl et al, 2015; Brembilla et al, 2018). Other cytokines are also involved in the polarization and amplification of T helper (Th)17/Th22 responses, such as IL-1 family members and more particularly IL-36 cytokines (Bassoy et al, 2018; Madonna et al, 2019).

The IL-36 cytokine family comprises three agonists, IL-36α, IL-36β, and IL-36γ, and an antagonist IL-36Ra. IL-36 agonists bind to a similar heterodimeric receptor complex, composed of the IL-36 receptor (IL-36R) and IL-1RAcP co-receptor (Bassoy et al, 2018; Madonna et al, 2019). Despite being physiologically present in healthy skin, mostly in epidermal compartment, IL-36α and IL-36γ are markedly up-regulated in psoriatic lesions (Blumberg et al, 2007; Carrier et al, 2011; Bassoy et al, 2018; Madonna et al, 2019), in which they are produced in high amounts by keratinocytes, but also, in a lesser extent, by dermal fibroblasts, monocytes, macrophages, dendritic cells, or possibly neutrophils (Bachmann et al, 2012; Bassoy et al, 2018; Madonna et al, 2019). IL-36R is expressed in different cell types present in psoriatic lesions, such as keratinocytes, endothelial cells, or Langerhans cells (Dietrich et al, 2016; Bridgewood et al, 2017). In keratinocytes, IL-36 stimulation promotes the release of chemokines and pro-inflammatory cytokines involved in psoriasis pathogenesis, notably in recruitment of neutrophils, and interferes with the processes of differentiation and cornification (Foster et al, 2014; Li et al, 2014; Bridgewood et al, 2017; Scheibe et al, 2017).

A large body of evidence suggests that IL-36 cytokines exert a pathogenic role in psoriasis. Aldara cream containing imiquimod, a TLR7 agonist, triggers the development of psoriasis-like skin dermatitis (van der Fits et al, 2009). IL-36α- (but not IL-36β- or IL-36γ-) deficient mice exhibited reduced skin pathology in response to Aldara application (Milora et al, 2015). On the other hand, IL-36Ra-deficient mice developed aggravated Aldara-induced pathology (Tortola et al, 2012). Importantly, defective IL-36 signaling in radio-resistant cells was shown to be essential for the development of Aldara-induced psoriasis-like dermatitis (Tortola et al, 2012).

Because of the important expression of IL-36R in keratinocytes and their role in production of pro-inflammatory molecules in response to IL-36 stimulation, we hypothesized that IL-36R signaling in keratinocytes is key for the development of Aldara-induced psoriasis-like skin dermatitis. Herein, we show that $K5Cre \times Il36r^{fl/fl}$ mice with a specific deletion of IL-36R in keratinocytes exhibited a similar resistance to Aldara-induced psoriasis-like dermatitis as IL-36R deficient mice.

---

[1]Department of Pathology-Immunology, University of Geneva Faculty of Medicine, Geneva, Switzerland [2]Division of Rheumatology, Department of Medicine, University Hospitals of Geneva, Geneva, Switzerland

Correspondence: cem.gabay@unige.ch
Jennifer Palomo's present address is Université Paris-Saclay, Institut National de Recherche pour l'Agriculture, l'Alimentation et l'Environnement (INRAE), AgroParisTech, Micalis Institute, Jouy-en-Josas, France

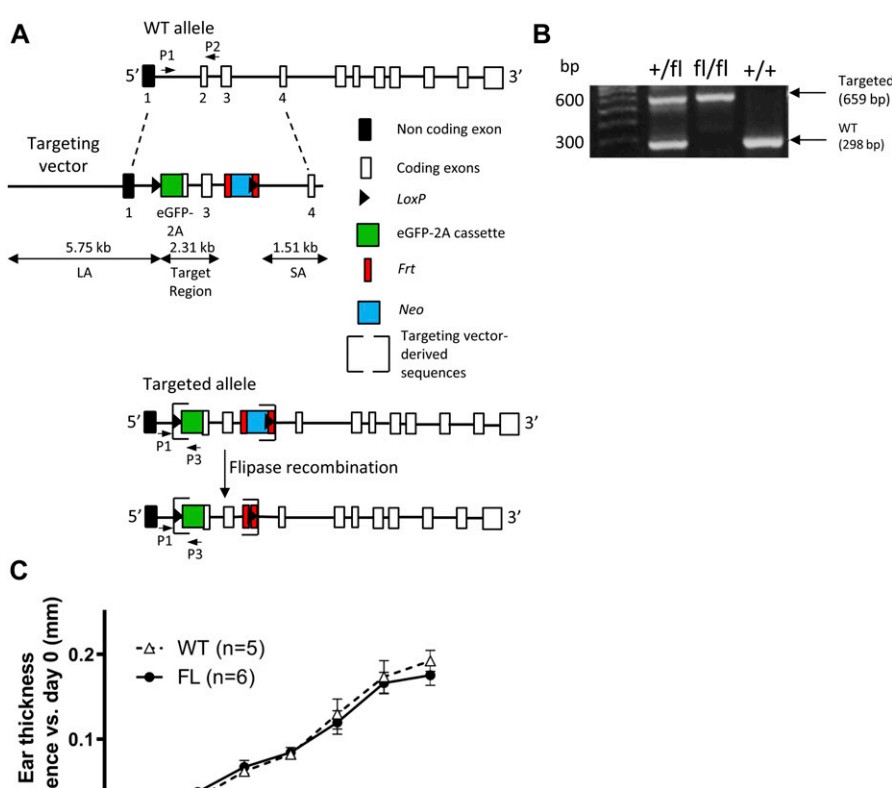

**Figure 1. Generation of *Il36r*-floxed (*Il36r*fl/fl) mice and characterization of their susceptibility to Aldara-induced skin inflammation.**
**(A)** Schematic representation of wild-type (WT) and *Il36r* mutated alleles is shown. Targeting strategies used for insertion of the *eGFP* reporter gene and conditional deletion of the mouse *Il36r* exons 2 and 3 are depicted. From top to bottom represents the WT *Il36r* genomic locus (WT allele), the targeting vector, targeted allele, and *Flp* recombinase-mediated excision of neomycin (Neo) resistance gene (*Il36r*-floxed allele). Coding exons are represented as white boxes and noncoding exon as black box. Targeting vector-derived sequences are indicated between brackets. Exons 2–3 were replaced by the following construct: *LoxP* (triangle)–eGFP-2A peptide (green box)–Exon2–Exon 3–*Frt* (red stripe)–*Neo* (blue box)–*Frt* (red stripe)–*LoxP* (triangle). Location of primers designed for genotyping (P1 and P2 for WT allele, P1 and P3 for targeted allele) is indicated. **(B)** PCR products in homozygous floxed (fl/fl), heterozygous floxed (+/fl), and WT mice (+/+) are shown. **(C)** Female *Il36r*fl/fl (FL) mice and WT littermates were challenged with the topical application of Aldara during 7 d. Statistical analysis was performed with two-way ANOVA followed by Holm–Sidak's comparison. A *P*-value < 0.05 was considered as significant.

# Results

## *Il36r*fl/fl and K5Cre, but not K14Cre mice, are sensitive to Aldara-induced psoriasis-like skin inflammation

The targeting construct used to generate *Il36r*fl/fl (FL) mice is depicted in Fig 1A and described in details in the Materials and Methods section. FL mice were fertile and healthy. However, unfortunately, the eGFP reporter which should have allowed tracking of IL-36R expression could not be detected, neither by FACS or immunofluorescence (data not shown).

To make sure that FL mice are sensitive to Aldara-induced psoriasis-like skin dermatitis, we treated them and their wild-type (WT) littermates daily with Aldara cream for 7 days (d) on one ear, whereas the other ear was left untreated, as control. WT littermates showed a continuous increase in ear thickness over the 7 d of treatment (Fig 1C) (Tortola et al, 2012; Palomo et al, 2018). Importantly, both FL and K5*Cre* mice presented a similar increase in ear thickness over the course of the treatment as their WT littermates (Fig 1C). To specifically delete *Il36r* expression in keratinocytes, we could cross FL mice with either K5*Cre* or K14*Cre* mice, in which the Cre recombinase is expressed under the control of K5 or K14 promoters, respectively, both markers of basal keratinocytes (Coulombe et al, 1989). These two mice strains were treated daily with Aldara cream, as described before. Although K5*Cre* mice

presented similar sensitivity to the treatment than their WT littermate controls (Fig S1A), this was not the case for K14*Cre* mice, which were partly resistant to the increase in ear thickness induced by Aldara treatment (Fig S1B). Thus, we selected K5*Cre* mice for further experiments.

## Specific deletion of IL-36R in keratinocytes of Il36rΔK mice

To generate mice with specific deletion of IL-36R in keratinocytes, we crossed FL and K5*Cre* mice to obtain keratinocyte-specific IL-36R knockout (termed *Il36r*ΔK, DK) mice. These mice were healthy, fertile, and presented similar body weight as their FL littermates (Fig 2A).

We first measured Il1rl2 mRNA expression levels (coding for IL-36R) in different organs and compared them with those of their FL littermate control mice. We observed that Il1rl2 mRNA expression levels were relatively low in lung and heart but similar in FL and DK mice (Fig 2B). Importantly, Il1rl2 mRNA levels were elevated in ears of FL mice but almost abolished in the ears of DK mice (Fig 2B), suggesting that IL-36R expression was successfully deleted in keratinocytes from DK mice.

To demonstrate the specific deletion of IL-36R in keratinocytes from DK mice, we cultured primary keratinocytes from FL and DK mice and measured Il1rl2 mRNA levels in these cells. BMDCs were used as controls because these cells express Il1rl2 mRNA and respond to IL-36 stimulation (Vigne et al, 2011, 2012). Il1rl2 mRNA

expression was entirely lacking in keratinocytes (Fig 2C) but was unaltered in BMDCs (Fig 2D) from DK mice, as compared with their FL littermates.

We then assessed the ability of keratinocytes and BMDCs to respond to IL-36 stimulation. We observed that keratinocytes from FL mice responded to IL-36$\beta$ by producing CXCL1, whereas IL-36$\beta$ was devoid of any stimulatory effect in DK keratinocytes (Fig 2E). Of note, keratinocytes from both FL and DK mice produced CXCL1 in response to IL-1$\beta$ stimulation, confirming that IL-1 signaling was unaltered in DK keratinocytes (Fig 2E). Importantly, IL-6 production in response to LPS and IL-36$\beta$ stimulation was similar in BMDCs from both FL and DK mice (Fig 2F). These results were further confirmed by measuring the expression of some genes associated with an IL-36 signature in human keratinocytes (Li et al, 2014; Mahil et al, 2017; Muller et al, 2018; Swindell et al, 2018) (Fig 2C and D). Taken together, these results showed that IL-36R expression and signaling were specifically suppressed in keratinocytes of DK mice.

### Il36r$^{\Delta K}$ mice are resistant to Aldara-induced psoriasis-like skin inflammation

To assess the importance of IL-36 signaling in keratinocytes for the development of Aldara-induced psoriasis-like dermatitis, we treated DK mice and their FL littermates with Aldara cream. As additional controls, we also treated Il36r$^{-/-}$ (KO) mice, fully deficient for IL-36R, and their Il36r$^{+/+}$ (WT) littermates with Aldara cream. The thickness of untreated ears remained unchanged and similar in the four mouse groups (Fig 3A). On the contrary, the thickness of Aldara-treated ears significantly increased during the 7-d treatment course in both WT and FL mice, whereas both KO and DK mice were significantly and similarly resistant to Aldara-induced psoriasis-like dermatitis (Fig 3B), demonstrating that IL-36 signaling in keratinocytes is mandatory for the development of Aldara-induced psoriasis-like disease.

### Keratinocyte-specific deletion of IL-36R does not prevent increased epidermis thickness in Aldara-induced psoriasis-like skin inflammation

To further investigate the protected skin phenotype conferred by the specific deletion of IL-36R in keratinocytes, hematoxylin and eosin (H&E) staining was performed on tissue sections from Aldara-treated or untreated ears at d7 of treatment (Figs 3C and S2). We could confirm that Aldara-treated ears from WT and FL mice presented symptoms of psoriasis, with increased cellular infiltrate in the dermis (Fig 3C) and epidermis thickness (acanthosis) as compared with untreated ears (Fig S2). These features were markedly reduced in Aldara-treated ears from KO mice (Fig 3C and D). In contrast, the epidermis thickness was not reduced in DK compared with FL mice (Fig 3C and D), indicating that keratinocyte hyper-proliferation is independent of IL-36 signaling in keratinocytes.

### Impairment of IL-36 signaling in keratinocytes abolishes Aldara-induced neutrophil recruitment in the ear

To characterize how the lack of IL-36 signaling in keratinocytes affects psoriasis-like dermatitis and the distribution of CD45$^+$

subsets, FACS analysis was performed in Aldara-treated or untreated ears, draining and non-draining lymph nodes, spleen, and blood of WT, KO, FL, and DK mice at d3 and d7.

To exclude blood CD45$^+$ cells from analyzed tissues, fluorochrome-labeled anti-CD45 antibody was injected iv 2 min before euthanizing. Almost all blood cells were stained with iv-injected anti-CD45 antibody (Fig S3). Blood CD45$^+$ cells represented in average 10–40% of CD45$^+$ cells of Aldara-treated ears and around 40–60% in untreated ears, whereas the frequency of these cells was lower than 10% of total CD45$^+$ cells in lymph nodes (Fig S3), as previously observed (Anderson et al, 2014). All further analyses were performed by excluding blood CD45$^+$ cells.

Using t-SNE multidimensional FACS analysis, we characterized the distribution of CD45$^+$ cell subsets in Aldara-treated ears at d3 and d7 (Fig 4A). We observed that the dominant subset at d3 in IL-36R–competent WT and FL mice was by far Ly6G$^+$ cells, representing around 50% of tissue CD45$^+$ cells (Fig 4B). The frequencies of Ly6G$^+$ cells were significantly reduced in both KO and DK mice at d3 (Fig 4B). At d7, the frequencies of Ly6G$^+$ cells were reduced compared with d3 in WT and FL mice, but still significantly higher than in KO and DK mice, respectively (Fig 4B).

The other dominant cell subsets among CD45$^+$ were B220$^+$, CD4$^+$, and CD8$^+$ T cells, with frequencies of around 5–15% for each of these subsets in WT and FL mice, and around 15–30% in KO and DK mice at d3 (Fig 4B).

To better characterize the composition of CD45$^+$ cells, especially rare subsets that could not be precisely defined by t-SNE analysis, we performed a classical gating strategy defined in Fig S4A.

We observed an increase in absolute numbers of CD45$^+$ cells, notably B cells, plasmacytoid dendritic cell-enriched subset (pDC), TCR$\gamma\delta^+$, CD4$^+$, CD8$^+$ T cells, innate lymphoid cell-enriched subset (ILC), monocytes, and neutrophils at d3 and d7 in Aldara-treated compared with untreated ears in all mouse lines (Fig 5).

Neutrophil numbers were markedly reduced in both KO and DK mice compared with their WT and FL littermates, at both d3 and d7 (Fig 5). Of note, and as previously observed (Tortola et al, 2012), absolute numbers of neutrophils were reduced at d7 compared with d3 in both WT and FL mice (Fig 5). Absolute numbers of ILCs were also significantly reduced in both KO and DK mice compared with their littermates at d3 (Fig 5).

Numbers of pDCs (at d3 and d7), TCR$\gamma\delta^+$ (at d3 and d7), and monocytes at d3 were reduced in KO mice compared with WT mice but were not significantly different between FL and DK mice in Aldara-treated ears (Fig 5).

In dLNs, absolute numbers of TCR$\gamma\delta^+$ and CD4$^+$ T cells increased at d7 as compared with d3. We observed reduced numbers of TCR$\gamma\delta^+$ T cells in KO mice compared with WT mice at d7 but not at d3 (Fig S4B). Thus, IL-36 signaling in other cells than in keratinocytes, might regulate numbers of TCR$\gamma\delta^+$ T cells in dLN after Aldara treatment.

Taken together, our results demonstrate that whereas pDC, monocyte, and TCR$\gamma\delta^+$ T cell numbers are regulated by global IL-36 signaling, neutrophil and ILC numbers are selectively regulated by IL-36 signaling in keratinocytes in Aldara-treated ears.

### Production of IL-23 at d3 of Aldara treatment is entirely dependent on IL-36 signaling in keratinocytes

To characterize the genes specifically regulated by keratinocyte-dependent IL-36 signaling (KC36), we performed RNASeq analysis

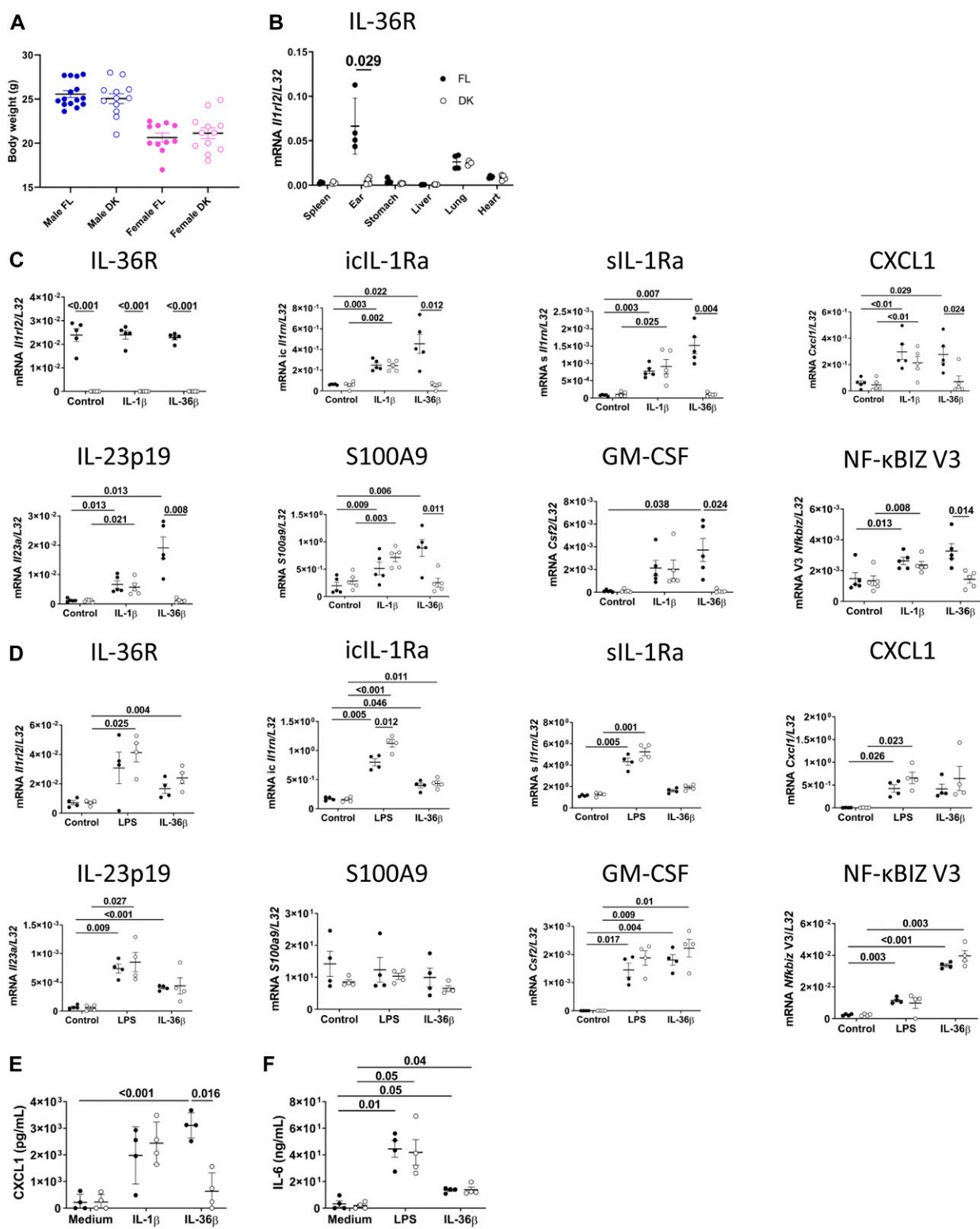

**Figure 2. IL-36R is specifically deleted in keratinocytes of *Il36r*^ΔK mice.**
**(A)** Comparison of body weights from 8- to 20-wk-old *Il36r*^fl/fl (FL) and *Il36r*^ΔK (DK) mice. **(B)** The indicated organs were collected from FL and DK mice, and levels of *Il1rl2* mRNA expression (coding for IL-36R) were determined by qRT-PCR. The results represent mRNA expression relative to L32. Results are from one representative experiment of two. **(C, E)** Keratinocytes were prepared from FL and DK mice and then cultured for 6 h with medium alone, murine IL-1β (100 ng/ml), or murine IL-36β (100 ng/ml). **(C)** Levels of mRNA expression of the genes coding for the indicated proteins in stimulated keratinocytes were determined by qRT-PCR. The results represent mRNA expression relative to *L32*. Results are pooled from five independent experiments. **(E)** The levels of CXCL1 were measured in the cell supernatants by ELISA.

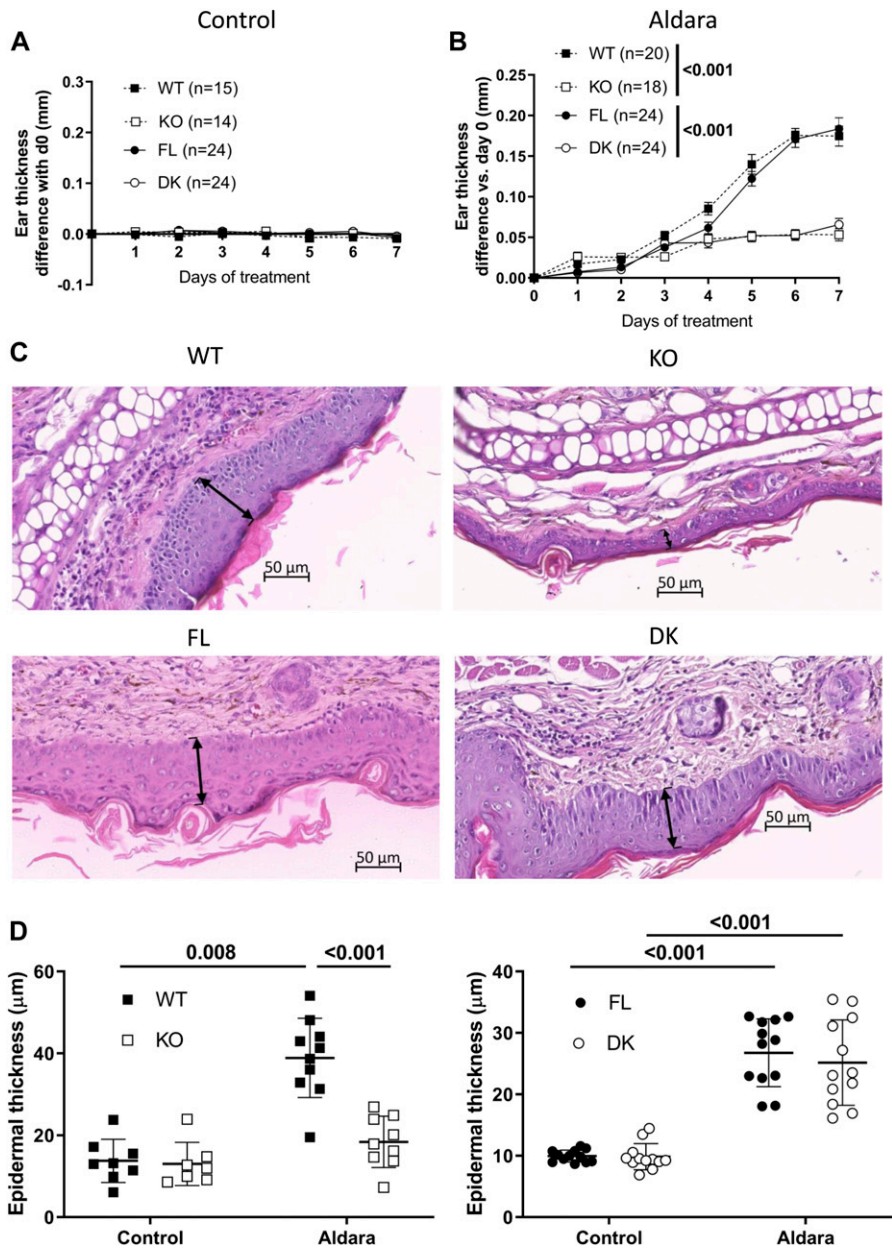

**Figure 3. _Il36r_<sup>ΔK</sup> mice are protected from Aldara-induced skin inflammation but not acanthosis.**
_Il36r_<sup>−/−</sup> (KO) and _Il36r_<sup>ΔK</sup> (DK) mice as well as their littermate controls, respectively, _Il36r_<sup>+/+</sup> (WT) and _Il36r_<sup>fl/fl</sup> (FL) mice, were challenged with the topical application of Aldara on one ear during 7 d. **(A, B)** The severity of skin inflammation was assessed by daily measurement of ear thickness in control (A) and Aldara-treated (B) ears. The number in parentheses indicates the number of mice for each genotype. Differences of ear thickness between WT and KO mice or between FL and DK mice are represented at the indicated days. Data are combined from 8 to 10 independent experiments. **(C)** Representative hematoxylin and eosin (H&E)–stained sections of four independent experiments from Aldara-treated ears of indicated mice at day 7 of treatment. The arrows highlight the thickness of epidermis. **(D)** Epidermal thickness was measured from H&E–stained sections of the ears of WT and KO mice (left panel) or FL and DK mice (right panel) after 7 d of Aldara treatment. Averages are calculated from 20 random measurements and combined from four independent experiments. Each dot represents the average value obtained from one mouse. **(A, B, C, D)** Statistical analysis was performed with two-way ANOVA followed by Holm–Sidak's comparison (A, C, B), and Wilcoxon and Mann–Whitney tests (D). Numbers represent the _P_-values when significance was found. A _P_-value < 0.05 was considered as significant.

on Aldara-treated and untreated ears of WT, KO, FL, and DK mice at d3 and d7. The strategy used to define and validate IL-36–dependent genes and KC36 genes is described in Figs S5 and S6. We identified 214 and 675 IL-36–dependent genes at d3 (Fig S5A and Table S1) and d7 (Fig S5B and Table S1), respectively, which included 5 and 150 KC36-dependent genes at d3 (Fig S6A and Table S1) and d7 (Fig S6B and Table S1), respectively. The top 30

up- and down-regulated genes of each category are depicted in Fig 6.

Because we could obtain only 5 KC36-dependent genes at d3 following our strategy of selection, we reasoned that the threshold of selection might have been too narrow. To further validate these results and expand the threshold, we analyzed the top 30 selected genes from each category based on the comparison of reads per

**(D, F)** BMDCs were prepared from _Il36r_<sup>fl/fl</sup> (FL) and _Il36r_<sup>ΔK</sup> (DK) mice. BMDCs were cultured for 24 h with medium alone, LPS (10 ng/ml), or murine IL-36β (100 ng/ml). **(D)** Levels of mRNA expression of the genes coding for the indicated proteins in stimulated BMDCs were determined by qRT-PCR. The results represent mRNA expression relative to L32. Results are pooled from two independent experiments. **(F)** Levels of IL-6 were measured in the cell supernatants by ELISA. **(B, C, D, E, F)** Statistical analysis was performed using the Mann–Whitney test (B), RM one-way ANOVA, and Welch _t_ tests (C, D, E, F). Numbers indicated represent the _P_-values when significance was found. A _P_-value < 0.05 was considered as significant.

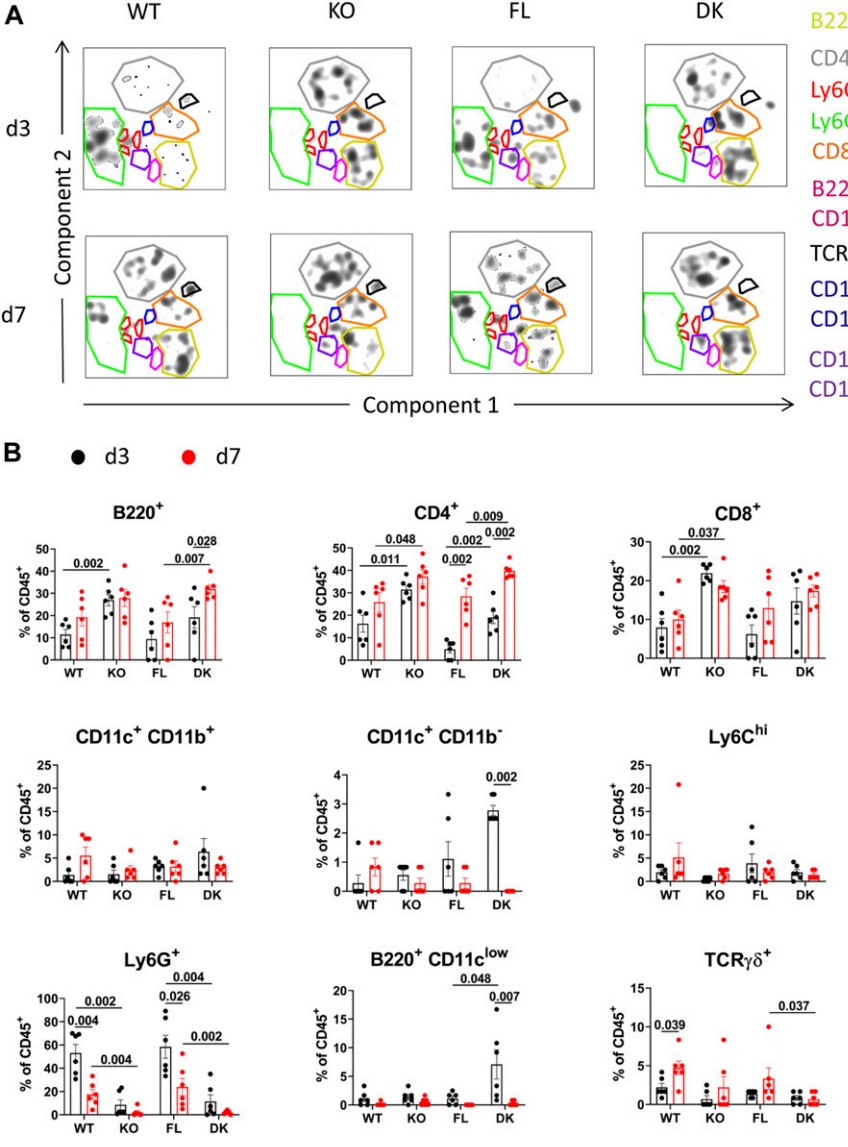

**Figure 4. Ly6G⁺, B220⁺, CD4⁺, and CD8⁺ cells are dominant among tissue CD45⁺ cells from Aldara-treated ear.**

*Il36r⁻/⁻* (KO) and *Il36r*^ΔK (DK) mice as well as their littermate controls, respectively, *Il36r⁺/⁺* (WT) and *Il36r*^fl/fl (FL) mice, were challenged with the topical application of Aldara on one ear during 3 or 7 d. **(A)** Multidimensional analysis (t-SNE) by FACS of the live CD45 iv⁻ CD45⁺ cell subsets from Aldara-treated ears at day 3 (d3) or day 7 (d7) of treatment. The most represented subsets are highlighted with color gates, and legend is indicated on the right of the panels. **(B)** The graphs depict frequencies among live CD45⁺CD45 iv⁻ of the indicated subsets from multidimensional analysis (t-SNE) from Aldara-treated ears at day 3 or 7. Results are combined from six independent experiments. Statistical analysis was performed with Mann–Whitney test. Numbers represent the *P*-values when significance was found. A *P*-value < 0.05 was considered as significant.

kilobase million (RPKM) raw data values. The list of confirmed genes from each category is shown in Table 1, and information on their functions is shown in Table S2. We also performed qRT-PCR validation on a panel of pro-inflammatory genes up-regulated at d3 and d7 (Fig S7).

IL-36α, IL-36β, and IL-36γ did not belong to the top 30 up-regulated genes, at neither d3 nor d7. However, all of them were up-regulated in Aldara-treated ears (Fig S8A and B and Table S1). Both IL-36α and IL-36γ were regulated in an IL-36–dependent, KC36-independent manner at d3 and d7, whereas IL-36β was regulated in an IL-36–dependent, KC36-independent manner only at d7 and was IL-36–independent at d3 (Fig S8A and B). Thus, our results demonstrated that IL-36 cytokines partly regulate their own production in Aldara-treated ears.

We observed a differential pattern of gene expression regulated in a KC36-dependent manner at d3 and d7. Indeed, at d3, key pro-inflammatory genes of the IL-23/IL-17/IL-22 axis (Il22, Il12b,

Il17f, and Il23a) and neutrophil-associated genes (Ly6g, Csf3, Cxcl3, Il17f, and Cd300lf) were up-regulated in Aldara-treated ears (Tables 1, S1, S2, and Fig S7). Of note, both fold change and RPKM analysis indicated that Il23a and Il12b transcripts, coding for IL-23 subunits, were up-regulated in a KC36-dependent manner in Aldara-treated ears at d3 (Fig 6 and Table 1). The role of KC36-dependent signaling on IL-23p19 expression was confirmed by qRT-PCR analysis on Aldara-treated ears as well as on cultured keratinocytes (Figs 2 and S7). At d7, we observed that expression of cytokines of the IL-23/IL-17/IL-22 axis (Il17a, Il12b, Il17f, and Il17c) was also regulated by IL-36R signaling, but in a KC36-independent manner (Tables 1 and S1).

It was recently demonstrated that Il23p19 heterodimerizes with EBI3 in human keratinocytes to form the new cytokine IL-39 and can present IL-23–independent functions (Ramnath et al, 2015). Thus, we assessed Ebi3 regulation in Aldara-treated ears using RNASeq analysis. Ebi3 was found up-regulated in Aldara-treated ears in an

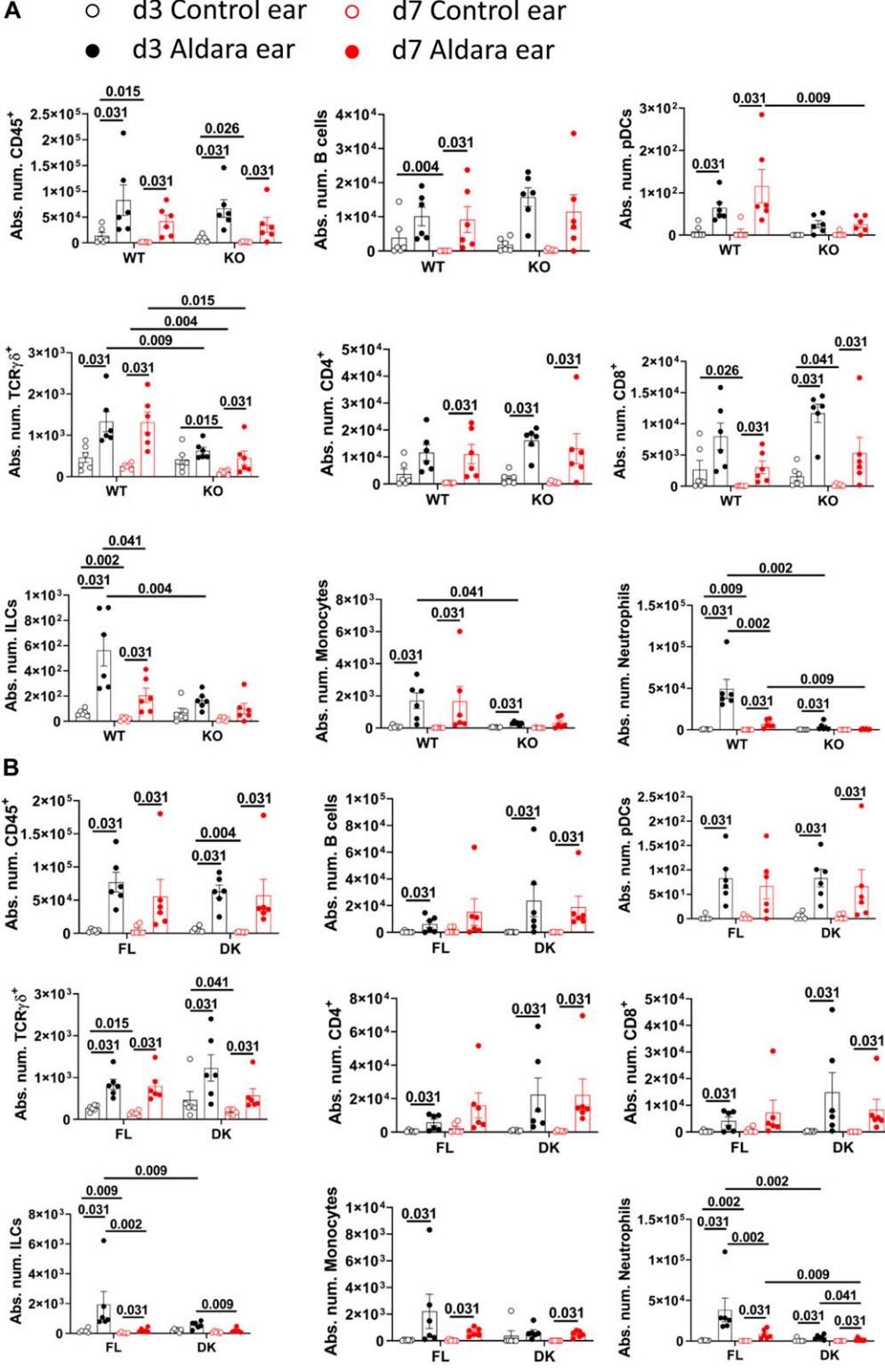

**Figure 5.  Impairment of IL-36 signaling in keratinocytes abolishes Aldara-induced neutrophil recruitment in the ear.**

*Il36r*[-/-] (KO) and *Il36r*[ΔK] (DK) mice as well as their littermate controls, respectively, *Il36r*[+/+] (WT) and *Il36r*[fl/fl] (FL) mice, were challenged with the topical application of Aldara on one ear during 3 or 7 d. **(A)** Absolute numbers of the indicated cell subsets (defined in Fig S4A) from Aldara-treated ears of WT and KO mice at the indicated time points. **(B)** Absolute numbers of the indicated cell subsets (defined in Fig S4A) from Aldara-treated ears of FL and DK mice at the indicated time points. Results are combined from six independent experiments. Statistical analysis was performed with Wilcoxon and Mann–Whitney tests. Numbers represent the *P*-values when significance was found. A *P*-value < 0.05 was considered as significant.

IL-36–independent manner at d7 (Fig S8C and Table S1). Importantly, Ebi3 expression could not be detected at d3, which corresponds to the IL-36–dependent KC36-dependent regulation of both IL-23 subunits (Fig 6 and Table 1), suggesting that it is IL-23 and not IL-39 which exerts functions at d3 of Aldara-induced psoriasis-like dermatitis.

Up-regulated KC36-dependent genes at d7 were enriched in genes associated with myeloid cells, especially with antigen-presenting cell

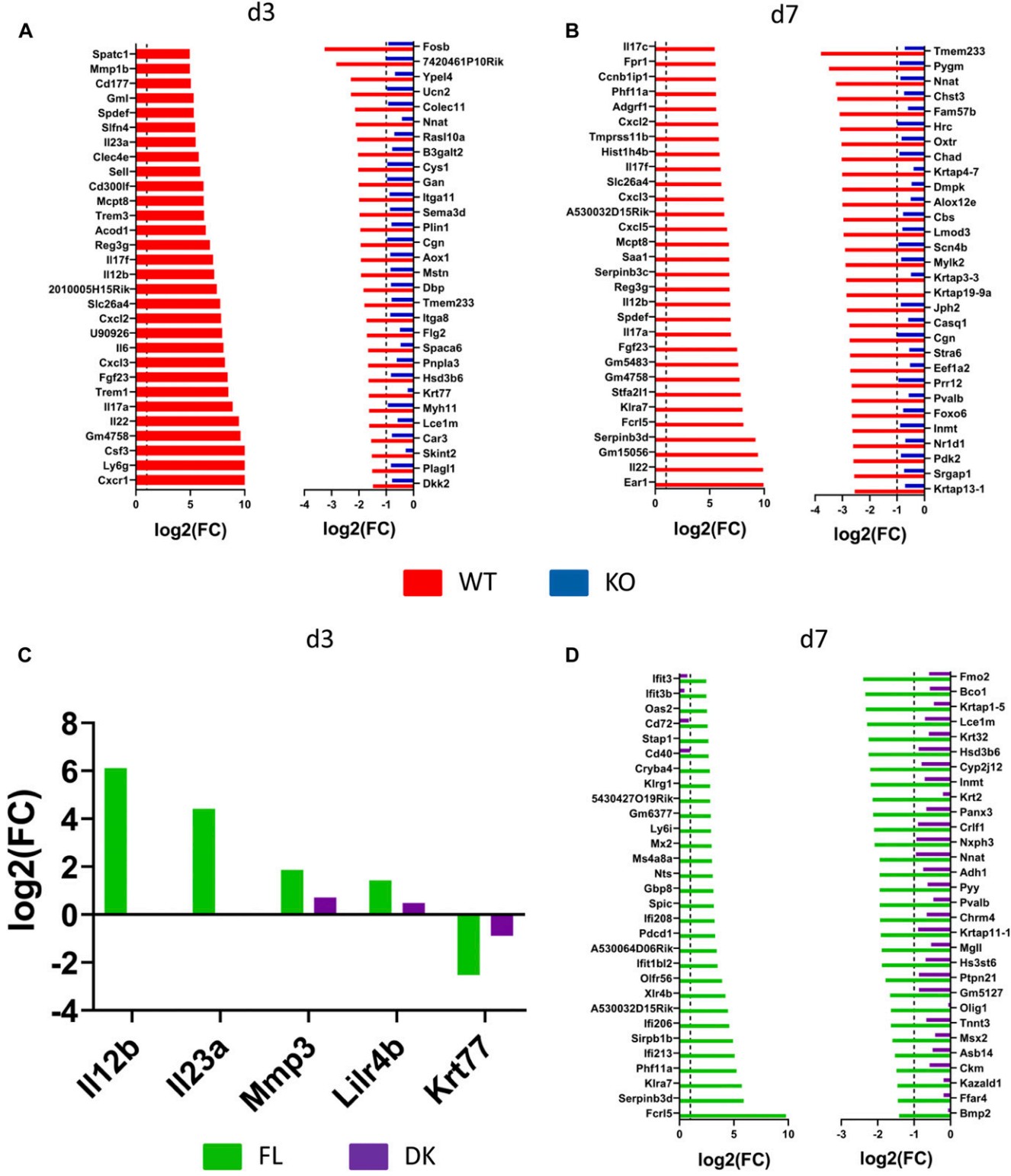

**Figure 6. IL-23 expression at day 3 is regulated by IL-36R signaling in keratinocytes after Aldara treatment.**
*Il36r*[−/−] (KO) and *Il36r*[ΔK] (DK) mice as well as their littermate controls, respectively, *Il36r*[+/+] (WT) and *Il36r*[fl/fl] (FL) mice, were challenged with the topical application of Aldara on one ear during 3 or 7 d. RNASeq was performed on treated and untreated ears. **(A, B)** These graphs represent the log$_2$ of the fold changes of gene expression of the 30 most up-regulated and down-regulated genes among IL-36–dependent genes (as defined in Fig S5) in Aldara-treated compared with untreated ears at d3 (A) and d7 (B) of treatment. **(C, D)** This graph represents the log$_2$ of the fold changes of gene expression of the keratinocyte IL-36R (KC-36)–dependent genes (as defined in Fig S6) in Aldara-treated compared with untreated ears at d3 (C) and d7 (D) of treatment. Results are combined from five independent experiments.

**Table 1. List of RNASeq genes validated by analysis of RPKM.**

| Conditions | Up-regulated genes | Down-regulated genes |
|---|---|---|
| IL-36 dependent, KC36 dependent, d3 | Ly6g, Csf3, Il22, Cxcl3, U90926, Cxcl2, Slc26a4, Il12b, Il17f, Acod1, Cd300lf, Il23a, Slfn4, Spdef, Mmp3, and Lilr4b | Rasl10a, Plin1, and Car3 |
| IL-36 dependent, KC36 independent, d3 | Cxcr1, Gm4758, Il17a, Trem1, Fgf23, Il6, 2010005H15Rik, Reg3g, Trem3, Mcpt8, Sell, Clec4e, Gml, Cd177, Mmp1b, and Spatc1 | Ypel4, Colec11, Nnat, B3galt2, Cys1, Itga11, Sema3d, Cgn, Aox1, Dbp, Tmem233, Itga8, Flg2, Pnpla3, Hsd3b6, Krt77, and Lce1m |
| IL-36 dependent, KC36 dependent, d7 | Serpinb3d, Serpinb3c, A530032D15Rik, Hist1h4b, Fpr1, Olfr56, Ms4a8a, Ly6i, Gm6377, Sirpb1b, Xlr4b, A530064D06Rik, Klrg1, Pdcd1, and Spic1 | Fam57b, Oxtr, Alox12e, Foxo6, Chrm4, Olig1, Kazald1, Ffar4, Fmo2, Lce1m, Crlf1, Msx2, Cyp2j12, Krt2, Bmp2, Hsd3b6, Panx3, Nxph3, Mgll, Hs3st6, Ptpn21, Gm5127, and Tnnt3 |
| IL-36 dependent, KC36 independent, d7 | Ear1, Il22, Stfa2l1, Gm4758, Gm5483, Il17a, Spdef, Il12b, Saa1, Cxcl5, Cxcl3, Slc26a4, Il17f, Tmprss11b, Cxcl2, Adgrf1, Il17c, and Nts | Tmem233, Pygm, Nnat, Chst3, Hrc, Dmpk, Cbs, Lmod3, Scn4b, Mylk2, Krtap3-3, Jph2, Casq1, Cgn, Stra6, Eef1a2, Pvalb, Inmt, Nr1d1, Pdk2, Srgap1, Bco1, Krt32, Krtap11-1, Adh1, Asb14, and Ckm |

function (Fpr1, Ms4a8a, Ly6i, Sirpb1b, Xlr4b, and A530064D06Rik), whereas down-regulated genes were associated with skin homeostasis, wound healing, and keratinocyte function (Fam57b, Oxtr, Alox12e, Chrm4, Kazald1, Ffar4, Fmo2, Lce1m, Msx2, Cyp2j12, Krt2, Bmp2, Hsd3b6, Panx3, Nxph3, Hst3st6, Ptpn21, Gm5127, and Tnnt3) (Tables 1 and S2).

In addition, an important fraction of the validated genes were involved in keratinocyte proliferation and/or differentiation and were modulated both in a KC36-dependent and KC36-independent manner at d3 and d7. These genes are Il22, Il17f, Mmp3, Car3 (KC36-dependent, d3), Il17a, Il6, Reg3g, Mmp1b, Nnat, Sema3d, Cgn, Dbp, Flg2, Krt77, and Lce1m (KC36-independent, d3); Serpinb3d, Fam57b, Oxtr, Alox12e, Chrm4, Kazald1, Ffar4, Fmo2, Lce1m, Msx2, Krt2, Bmp2, Panx3, Hs3st6, Ptpn21, Tnnt3 (KC36-dependent, d7); and Ear1, Il22, Gm5483, Il17a, Saa1, Il17f, Tmprss11b, Il17c, Nnat, Cbs, Krtap3-3, Cgn, Stra6, Krt32, Krtap11-1, and Adh1 (KC36-independent, d7) (Tables 1 and S2). Importantly, Krt16, a marker of hyperproliferation of keratinocytes in psoriasis (Leigh et al, 1995) did not belong to the top 30 up-regulated genes but was still up-regulated in an IL-36–dependent, KC36-independent manner in Aldara-treated ears at d7 (Fig S9C and Table S1). Its expression was not detected at d3 (data not shown).

Type I interferons produced by pDCs constitute another pathway of crucial importance for initiation of acute psoriasis in humans (Nestle et al, 2005; Conrad & Gilliet, 2018). Because pDC numbers were regulated in Aldara-treated ears in an IL-36–dependent manner at d3, we assessed expression of type-I interferon-related genes in Aldara-treated ears using RNASeq analysis. We identified some type-I interferon-related genes (from gene ontology biological processes database) among Aldara-induced genes: Tmem173, Cd14, Zbp1, Pycard, Traip, and Rnf125 at d3 and Nmi, Ptpn22, Dhx58, Zbp1, Ifitm3, and Irf7 at d7. At d3, Tmem173, Cd14, and Zbp1 were regulated in an IL-36–dependent, KC36-independent manner, whereas Pycard, Traip and Rnf125 were not regulated by IL-36 (Fig S9A and data not shown). At d7, Ptpn22 and Ifitm3 were regulated in an IL-36–dependent, KC-36-dependent manner, whereas Nmi, Dhx58, Zbp1, and Irf7 were not regulated by IL-36 (Fig S9B and data not shown). Importantly, neither Ifna nor Ifnb expression could be detected among Aldara-modulated genes at d3 or d7 (Table S1).

## Discussion

In the last decade, IL-36 cytokines have been closely associated with psoriasis. IL-36α, IL-36β, IL-36γ, and IL-36Ra expression levels are increased in psoriatic skin lesions (Debets et al, 2001; Blumberg et al, 2007; Johnston et al, 2011; Quaranta et al, 2014; D'Erme et al, 2015). A causative mutation of IL-36Ra has been associated with generalized pustular psoriasis (GPP) (Marrakchi et al, 2011; Onoufriadis et al, 2011) and treatment with a monoclonal antibody targeting IL-36R showed clinical efficiency in seven GPP patients (Bachelez et al, 2019). In transgenic mice, overexpression of IL-36α in keratinocytes induces spontaneous but transient skin inflammation, which is reactivated in adult mice in response to phorbol ester application (Blumberg et al, 2010). IL-36R–deficient mice are protected against Aldara-induced psoriasis-like dermatitis, and expression of IL-36R in radio-resistant cells is critical for the pathogenesis (Tortola et al, 2012). Our results show that specific deletion of IL-36R in keratinocytes is sufficient to confer almost similar protection to Aldara-induced psoriasis-like dermatitis as mice with a complete deletion of IL-36R, thus emphasizing the crucial role of IL-36R signaling in keratinocytes for the initiation of Aldara-induced skin pathogenesis.

Several studies highlighted the importance of early IL-36 production in psoriasis development. The antimicrobial peptide LL37, which induces type I IFN production in pDCs (Nestle et al, 2005; Lande et al, 2007; Kahlenberg & Kaplan, 2013), and *Staphylococcus aureus* epidermal colonization, which has been suggested to trigger psoriasis development via induction of Th17 cells (Chang et al, 2018), are both capable of inducing IL-36 production in keratinocytes (Nguyen et al, 2012; Li et al, 2014; Liu et al, 2017; Nakagawa et al, 2017). IL-36 cytokines then act on keratinocytes by amplifying their own production (Carrier et al, 2011; Bachmann et al, 2012; Mahil et al, 2017; Hashiguchi et al, 2018; Swindell et al, 2018). We could confirm here that expression of IL-36 cytokines was importantly induced in Aldara-treated ears and partially regulated by IL-36 itself. More particularly, IL-36α expression was induced around ten times more than the other IL-36 cytokines at d7, going along with a previous study which demonstrated using knockout animals that it is IL-36α but not IL-36β or IL-36γ, which is responsible of imiquimod-induced psoriasis (Milora et al, 2015). However, deleting specifically IL-36R

expression in keratinocytes was not sufficient to significantly reduce the induced expression of IL-36 cytokines, despite a marked tendency at d7. Thus, our results demonstrated that IL-36 signaling in keratinocytes plays only a minor role, if any, in IL-36 production during the course of Aldara-induced psoriasis-like dermatitis, despite the importance of IL-36 signaling in other cell types for IL-36 induction.

IL-36 cytokines also induce expression of several pro-inflammatory molecules in keratinocytes, such as neutrophilic chemo-attractants (CXCL1, CXCL2, and CXCL8) (Foster et al, 2014; Li et al, 2014; Hashiguchi et al, 2018), antimicrobial peptides (S100A7 and LL37) (Nguyen et al, 2012), and most importantly, IL-23p19 (Swindell et al, 2018), which stimulates the production of IL-17 and IL-22 by TCR$\gamma\delta^+$ or CD4$^+$ T cells (Stockinger & Veldhoen, 2007; Cai et al, 2011). We could confirm, here and in a recent study, that IL-36$\beta$ induces expression of IL-23p19 and CXCL1, but also of CXCL2, IL-36$\alpha$, IL-36$\gamma$, or G-CSF by keratinocytes (Martin et al, 2020). Most importantly, our RNASeq analysis allowed the distinction between genes induced upon IL-36R signaling in keratinocytes (KC36-dependent) from those dependent of IL-36R signaling in other cell types (KC36-independent), such as pDCs, monocytes, Langerhans cells, macrophages, dendritic cells, endothelial cells, or fibroblasts (Vigne et al, 2011, 2012; Foster et al, 2014; Dietrich et al, 2016; Bridgewood et al, 2017; Catapano et al, 2019; Madonna et al, 2019). Among the genes which were KC36 dependent, we found notably Cxcl2, Csf3 (coding for G-CSF), and Il23a (coding for IL-23p19) at d3.

IL-17, IL-23, and IL-22 cytokines have been closely correlated with IL-36 in human and Aldara-induced psoriasis (Carrier et al, 2011; D'Erme et al, 2015; Boutet et al, 2016). IL-17 and IL-22 induce IL-36 production in keratinocytes (Carrier et al, 2011; Pfaff et al, 2017) and blocking IL-36 attenuates the disease severity in a model of IL-23–induced psoriasis (Su et al, 2019). Another study performed in transgenic mice with keratinocytes expressing a constitutively active form of STAT3 showed that skin expression of Il23a was IL-36R dependent (Ohko et al, 2019). Our results demonstrate that IL-36R signaling in keratinocytes is crucial for IL-23, IL-17, and IL-22 production at d3 of Aldara treatment and suggest that IL-23 is produced by keratinocytes upon direct stimulation with IL-36.

IL-39 is a newly discovered cytokine, formed by the heterodimerization of IL-23p19 and Ebi3, common subunit with IL-27 and IL-35 (Wang et al, 2016). It has been shown that IL-23p19 heterodimerizes with EBI3 in human keratinocytes (Ramnath et al, 2015), suggesting that IL-23p19 expression might not be necessarily correlated with IL-23 function. In addition, it was recently observed that IL-36$\gamma$ could induce expression of both IL-23p19 and EBI3 in oral epithelial cells (Scholz et al, 2018). Thus, the IL-36–dependent, KC36-dependent increased expression of IL-23p19 that we observed at d3 could be instead associated with IL-39 and not IL-23 action. However, our RNASeq analysis revealed that Ebi3 is indeed up-regulated in Aldara-treated ears at d7 but not at d3, and that this up-regulation was independent of IL-36 signaling. On the other hand, Il12b and Il23a genes, coding for both subunits of IL-23, were up-regulated at d3 in an IL-36–dependent, KC36-dependent manner and were both comprised in the top 30 most up-regulated genes in Aldara-treated ears. Even if we cannot exclude the hypothesis that there was a local problem on Ebi3 detection by RNASeq analysis at d3, these results firmly suggest that it

is IL-23 and not IL-39 which is regulated in an IL-36–dependent, KC36-dependent manner at d3 of Aldara-induced psoriasis-like dermatitis.

Interestingly, at d7, production of IL-23/IL-17/IL-22–associated cytokines was IL-36 dependent but KC36 independent. Of note, although expression of Il23a could not be detected by RNASeq, qRT-PCR data showed that Il23a expression was dependent on IL-36R signaling in other cells than keratinocytes. These results indicate that production of Th17 cytokines in Aldara-treated ears follows two waves: the first wave is issued from IL-36R signaling in keratinocytes at d3, whereas IL-23 production is regulated at later time points (d7) by the stimulatory effect of IL-36 on other cell types (Vigne et al, 2011; Bridgewood et al, 2018).

Dermal TCR$\gamma\delta^+$ T cells were described as the main producers of IL-17 in the skin of mice with Aldara-induced psoriasis (Cai et al, 2011; Tortola et al, 2012). As previously shown (Tortola et al, 2012), the number of TCR$\gamma\delta^+$ T cells was reduced in KO compared with WT mice at d3 and d7 in comparison with FL mice. In contrast, ILC numbers were reduced in both KO and DK mice compared with their controls at d3 of Aldara treatment. Because ILC3 are also known as producers of IL-17A and IL-17F (Brembilla et al, 2018), one could hypothesize that KC36-mediated IL-23 production at d3 induces proliferation and production of IL-17 cytokines in skin ILC3 (Takatori et al, 2009). Interestingly, in contrary with some studies (Cai et al, 2011; Tortola et al, 2012), we observed lower numbers of TCR$\gamma\delta^+$ T cells in Aldara-treated skin and higher numbers of CD4$^+$ and CD8$^+$ T cells. Actually, our phenotype is closer to the one observed in psoriatic patients where skin T cells are dominated by TCR$\alpha\beta^+$ T cells (Matos et al, 2017) and of GPP patients whose clonally expanded IL-36–regulated CD4$^+$ T cells were the major producers of IL-17 (Arakawa et al, 2018). This observation suggests that the environment plays an important role in the representation of T-cell subsets, which may explain the differences of results regarding the proportion of TCR$\gamma\delta^+$ and TCR$\alpha\beta^+$ T cells.

It has been suggested that there exists three different signaling pathways that mediate three clinical phenotypes of psoriasis in patients: the chronic plaque-type psoriasis mediated by TNF, the pustular psoriasis by IL-36, supported by IL-1, cytokines, and the acute phase psoriasis mediated by type-I interferon (Conrad & Gilliet, 2018). More precisely, it was demonstrated in a xenograft model of uninvolved psoriasis skin that production of IFN-$\alpha$ by pDCs precluded, and was mandatory, for the onset of T-cell–mediated psoriasis (Nestle et al, 2005). However, despite the IL-36–mediated increase in pDC numbers at d3, we could not find expression of neither *Ifna* nor *Ifnb* at d3 or d7 of Aldara treatment. These results are in accordance with a previous study, in which the authors did not detect any increased production of IFN-$\alpha$ protein in the skin over a treatment course of 4 d, despite increased serum IFN-$\alpha$ and IFN-$\beta$ concentrations 3 h after the first Aldara application (Grine et al, 2015). Thus, these results and ours—including the mandatory role of IL-36 signaling, more particularly in keratinocytes, for development of the pathology—suggest that Aldara-induced psoriasis-like dermatitis is a better model for GPP than for acute phase psoriasis. This hypothesis is corroborated by another study using back skin treatment with another imiquimod-containing cream, which induces a more severe pathology than ear treatment only. In these mice, the authors observed systemic symptoms

associated with GPP in patients, such as anorexia, general malaise, and pain (Alvarez & Jensen, 2016).

Our findings also highlighted the critical role of IL-36R signaling in keratinocytes on the recruitment of neutrophils during the early stage of Aldara-induced psoriasis-like dermatitis. Neutrophils are found abundantly in psoriatic skin lesions (Chiang et al, 2019). We observed that neutrophils comprised up to half of the CD45$^+$ cells in WT and FL mice at d3, whereas their numbers significantly decreased at d7. Importantly, IL-36–dependent KC36-dependent top 30 genes at d3 were enriched in neutrophil-associated genes, such as surface molecules expressed by neutrophils (Ly6g and Cd300lf) or molecules involved in neutrophil trafficking (Csf3, Cxcl3, Cxcl2, and Il17f) (Brembilla et al, 2018; Chiang et al, 2019). Our results are also in line with the massive neutrophil infiltration that characterizes GPP (Chiang et al, 2019).

Early KC36-mediated neutrophil recruitment might be important for triggering an amplificatory feedback loop favoring IL-36 production and maturation. Indeed, IL-36 expression in inflamed skin is partly dependent on neutrophil extracellular trap activity (Shao et al, 2019), and neutrophil enzymes such as elastase or cathepsin G, present in neutrophil extracellular trap and released upon degranulation, activate IL-36 cytokines by cleavage of their N-terminal part and release of their mature, highly active, form (Johnston et al, 2017; Sullivan et al, 2018; Guo et al, 2019). Unfortunately, we were not able to detect either cathepsin G or elastase mRNA expression by RNASeq in Aldara-treated skin at d3 or d7, suggesting that these proteins might not be actively synthesized during the course of Aldara treatment.

KC36-dependent genes at d7 were enriched in genes associated with antigen-presenting myeloid cells (monocytes, macrophages, and different subsets of DCs) such as Fpr1, Ms4a8a, Ly6i, Sirpb1b, Spic1, Foxo6, Crlf1, Bmp2, Mgll, or Tnnt3. Neutrophil-associated genes were decreased (Fpr1 and Sirpb1b) at d7 compared with d3 among KC36-dependent genes, consistent with the important reduction of neutrophil infiltration and with a switch from innate to adaptive immunity in Aldara-induced psoriasis-like dermatitis (Tortola et al, 2012).

Deletion of IL-36R signaling in keratinocytes was not sufficient to prevent acanthosis in contrast to mice with complete deletion of IL-36R. A previous study corroborates our results by showing that acanthosis was similarly reduced in mice specifically deficient for the MyD88 (an element of IL-36 signaling pathway, [Bassoy et al, 2018]) in hematopoietic cells and in mice with complete deletion of MyD88 (Costa et al, 2017). Of note, and on the contrary to our results, the authors still observed significant acanthosis in MyD88-deficient mice compared with Vaseline-treated controls. However, they were using back skin treatment, which induces a much more severe form of psoriasis-like dermatitis and could explain the persistence of acanthosis in fully deficient animals. Anyhow, this result indicates that a functional MyD88 signaling in keratinocytes, such as IL-36 signaling, is not essential for acanthosis in Aldara-treated skin.

Importantly, Krt16 expression was also found up-regulated in an IL-36–dependent, KC36-independent manner at d7 but not at d3 in Aldara-treated ears. Because Krt16 expression is a marker of keratinocyte hyperproliferation in psoriasis (Leigh et al, 1995), this observation further corroborates the persistence of acanthosis observed in DK mice at d7.

This finding could be explained by the fact that expression of several genes associated with keratinocyte proliferation or altered differentiation were regulated in an IL-36–dependent but KC36-independent manner at both d3 (Il17a, Il6, Reg3g, Mmp1b, Nnat, Sema3d, Cgn, Dbp, Flg2, Krt77, and Lce1m) and d7 (Ear1, Il22, Gm5483, Il17a, Saa1, Il17f, Tmprss11b, Il17c, Nnat, Cbs, Krtap3-3, Cgn, Stra6, Krt32, Krtap11-1, and Adh1).

The genes of the IL-17/IL-22 axis fall in this category, as very important regulators of keratinocyte proliferation and differentiation. Indeed, IL-22 inhibits keratinocyte terminal differentiation and induces psoriasis pathogenic signature in keratinocytes (Hao, 2014). IL-17A also induces aberrant proliferation and altered differentiation of keratinocytes (Brembilla et al, 2018). IL-17C transgenic expression in keratinocytes induces psoriasis-like pathogenesis with thickening of the epithelium and production of pathogenic psoriasis cytokines (Johnston et al, 2013). Thus, the KC36-independent regulation of IL-17 cytokines at d7 suggests that these cytokines, among the other genes identified, induce acanthosis in DK mice.

While we were submitting this manuscript, another study performed with comparable methodology than ours demonstrated a similar resistance to Aldara-induced psoriasis-like dermatitis in K14*Cre* × *Il36r*$^{fl/fl}$ (DK14) mice than in *Il36r*-deficient mice, which corroborates our results (Hernandez-Santana et al, 2020). However, they observed some differences compared with our study. For instance, they found that acanthosis was prevented in DK14 mice compared with *Il36r*$^{fl/fl}$ mice after 6 d of treatment and a reduction in TCRγδ$^+$Vγ4$^+$IL-17A–producing T cells in Aldara-treated ears at d4 (Hernandez-Santana et al, 2020). However, we observed that K14*Cre* mice, but not K5*Cre* mice, were partly resistant to Aldara-induced psoriasis-like dermatitis. This difference may explain the difference between our data and those observed by Hernandez-Santana et al (2020).

Taken together, our results demonstrate that IL-36R signaling in keratinocytes is crucial for the development of Aldara-induced psoriasis-like dermatitis. Mechanistically, IL-36R signaling in keratinocytes is mandatory for IL-23 production and neutrophil infiltration at early time points.

# Materials and Methods

### Generation of Il36r$^{ΔK}$ and Il36r$^{fl/fl}$ mice

B6-*Il1rl2*$^{tm1iTL}$ (*Il36r*$^{fl/fl}$, FL) mice were generated by ingenious targeting laboratory. An 8.8-kb genomic DNA used to construct the targeting vector was first subcloned from a positively identified B6 BAC clone (RP23:216C5). The region was designed such that the long homology arm (LA) extends ~5.75 kb 5′ to the distal LoxP site and the short homology arm (SA) extends 1.51 kb 3′ to the flippase recognition target (FRT)-flanked Neo cassette. The eGFP-2A coding sequence was fused right after the endogenous adenine-thymine-guanine (ATG) translation initiation site in exon 2. The distal LoxP site was inserted 167-bp upstream of exon 2 within non-conserved sequence in rat predicted by Ensembl database. The FRT-flanked Neo cassette followed by a LoxP site was inserted downstream of

exon 3. The size of the target region is 2.31 kb containing exons 2–3 and the inserted eGFP-2A sequence (Fig 1A). The targeting vector was confirmed by restriction analysis and sequencing after each modification. The targeting vector was linearized by *NotI* and then transfected by electroporation of C57BL/6 (B6) embryonic stem cells. The targeted clones were identified by PCR and confirmed by Southern blot analysis. Targeted iTL IC1 (B6) embryonic stem cells were microinjected into Balb/c blastocysts. Resulting chimeras with a high percentage black coat color were mated to B6 FLP mice (B6.Cg-Tg(ACTFLPe) 9205Dym/J, The Jackson Laboratory [Jax]) to remove the Neo cassette. After Neo deletion, one LoxP-FRT site remains (ingenious targeting laboratory) (Fig 1A). Creation of the K5*Cre* transgenic lines was previously described (Tarutani et al, 1997). K5*Cre* mice (Tg(KRT5-cre)1Tak) were kindly provided by Dr Vincent Flacher (Institute of Cellular and Molecular Biology).

To induce a specific deletion of IL-36R in keratinocytes, these mice would be mated either with specific K5*Cre* or K14*Cre* lines, which express *Cre* recombinase under the control of the human keratin 5 or keratin 14 promoter, respectively, both markers of basal-layer keratinocytes. We selected K5*Cre* lines because K14*Cre* but not K5*Cre* mice showed intrinsic resistance to Aldara-induced psoriasis-like dermatitis (Fig S1B). Mating of K5*Cre* mice with *Il36r*$^{fl/fl}$ mice would, thus, induce specific Cre-mediated deletion in K5$^+$ keratinocytes in the progeny. Heterozygous progeny was then intercrossed and homozygous mice for the *Il36r*-floxed allele expressing Cre recombinase were obtained and named *Il36r*$^{\Delta K}$ mice.

This mouse line was maintained by intercrossing homozygous *Il36r*$^{fl/fl}$ female mice negative for the Cre-recombinase with homozygous *Il36r*$^{fl/fl}$ male mice expressing one allele of the Cre-recombinase transgene (*Il36r*$^{\Delta K}$). K14*Cre* mice (Jax #004782) were purchased from The Jackson Laboratory. WT C57BL/6J mice were obtained from Janvier. IL-36R–deficient mice (*Il36r*$^{-/-}$, KO) (B6.126S5-*Il1rl2*$^{tm1Hblu}$) obtained from Amgen Inc. (Blumberg et al, 2007) were backcrossed nine and seven times, respectively, in a pure C57BL/6J genetic background by using a marker-assisted selection protocol approach, also termed "speed congenics" (Lamacchia et al, 2007; Vigne et al, 2011). The *Il36r*$^{-/-}$ mouse line was maintained by intercrossing *Il36r*$^{+/-}$ mice. All mice were bred and maintained in the conventional animal facility of the Geneva University School of Medicine in a temperature-controlled (23°C) room with a 12-h light/dark cycle. Water and food were provided ad libitum. Animal studies were approved by the Animal Ethics Committee and the Geneva Veterinarian Office (licenses GE/71/17, GE/74/16, and GE/4/2019) and were performed according to the appropriate codes of practice.

### PCR genotyping

PCR was performed on genomic DNA purified from ear biopsies under the following conditions. Genotyping of *Il36r*$^{fl/fl}$ mice was performed using a three-primer PCR combining a common forward primer for the WT and targeted allele (5′-GCGTTTACTGGCTTCT-GAAGG-3′) with a reverse primer specific for the WT (5′-CAGAACT-TACCTGCTGCCACG-3′) or targeted (5′-GTCGATGCCCTTCAGCTCGATGCGG-3′) allele. Genotyping of *Il36r*$^{-/-}$ mice was performed using a three-primer PCR combining a common forward primer for the WT and targeted allele (5′-GCAGGCCAGTGAGAGAAAAGC-3′) with a reverse

primer specific for the WT (5′-CAGTCCAGGAAGCAAACACTGAAG-3′) or targeted (5′-TCTATCGCCTTCTTGACGAG-3′) allele. Genotyping of K5*Cre* mice was performed using a four-primer PCR combining two primers specific for the K5 human promoter; forward (5′-ATGCCAATGCCCCCT-CAGTTCCT-3′), reverse (5′-TGCCCCTTTTTATCCCTTCCAGA-3′) primers (Tarutani et al, 1997) and two primers specific for the WT; forward (5′-GAAAAGAGAGAGTGAATGGGAG-3′), reverse (5′-GAGCTCCATGATGTT-CACTGG-3′) primers. Genotyping of K14*Cre* mice was performed using a four-primer PCR combining two primers specific for the *Cre* sequence; forward (5′-GCGGTCTGGCAGTAAAAACTATC-3′), reverse (5′-GTGAAA-CAGCATTGCTGTCACTT-3′) primers and two primers specific for the WT; forward (5′-CTAGGCCACAGAATTGAAAGATCT), reverse (5′-GTAGGTG-GAAATTCTAGCATCATCC-3′) primers. The genomic DNA of *Il36r*$^{fl/fl}$ and K5*Cre* were amplified for 37 cycles with an annealing temperature of 57°C. Product sizes were 659 bp for the floxed and 298 bp for the WT allele (Fig 1B) in *Il36r*$^{fl/fl}$ mice. Product sizes were 299 bp for K5 promoter and 500 bp for the WT allele in K5*Cre* mice. K14*Cre* mice genomic DNA was amplified for 35 cycles with an annealing temperature of 51.7°C. Product sizes were 100 bp for the *Cre* and 324 bp for the WT allele. Genomic DNA of *Il36r*$^{-/-}$ mice was amplified for 35 cycles with an annealing temperature of 62°C. Product sizes were 610 bp for the targeted allele, 350 bp for the WT allele in *Il36r*$^{-/-}$ mice. *Il36r*$^{\Delta K}$ mice were genotyped with both protocols indicated above for *Il36r*$^{fl/fl}$ and K5*Cre* mice.

### Aldara-induced skin inflammation

Psoriasis-like skin disease was induced as reported previously (van der Fits et al, 2009). For Aldara-induced skin inflammation, 8–20-wk-old, age-matched *Il36r*$^{+/+}$, *Il36r*$^{-/-}$, *Il36r*$^{\Delta K}$, and *Il36r*$^{fl/fl}$ mice were challenged by daily topical application of 12.5 mg Aldara cream (5% imiquimod; MEDA Pharma GmbH) on the right ear for 3 or 7 d. Body weight was registered and ear thickness was assessed by daily measurement using a gauge (Mitutoyo Europe GmbH). For FACS experiments, mice received retro-orbital iv injection of 2 µg anti-CD45-BV421 (Clone 30-F11, 1/10; BD Biosciences) under isoflurane anesthesia. Then, after 2 min under anesthesia, mice received a supplementary intra-peritoneal injection of ketamine–xylazine anesthetic and were euthanized by exsanguination (cardiac puncture) followed by cervical dislocation. Ears were collected for histopathological evaluation, RNA extraction, or FACS staining at days 3 and 7 after the beginning of treatment.

### Antibodies

The following antimouse antibodies were used for FACS staining: anti-Ly6C-APC (clone AL-21, at a dilution of 1/100; BD Biosciences), anti NK1.1-Alexa Fluor 700 (clone PK136, 1/50; BD Biosciences), anti-CD4-PercP/Cy5.5 (clone RM4-5, 1/100; BD Biosciences), anti-CD8a-BV605 (clone 53-6.7, 1/200; BD Biosciences), anti-Ly6G-BV711 (clone 1A8, 1/400; BD Biosciences), anti-CD11c-BV786 (clone HL3, 1/100; BD Biosciences), anti-CD90.2-BV480 (clone 53-2.1, 1/200; BD Biosciences), anti-CD3e-BV650 (clone 145-2C11, 1/100; BD Biosciences), anti-CD45R/B220-BUV395 (clone RA3-6B2, 1/100; BD Biosciences), anti-CD11b-BUV737 (clone M1/70, 1/100; BD Biosciences), anti-CD16/CD32 purified (clone 2.4G2, 1/200; BD Biosciences), anti-γδ T-cell receptor-FITC (clone GL3, 1/100; BD Biosciences), anti-F4/80-PE

(clone T45-2342, 1/100; BD Biosciences), and anti-CD45-PE/Cy7 (clone I3/2.3, 1/100; BioLegend, Lucerna-Chem).

## Organ preparation and FACS staining

The spleen, blood, cervical, and mesenteric lymph nodes and both ears were collected from Aldara-treated mice.

Blood was collected with EDTA-coated syringes in 15-ml tubes containing 10 ml of cold FACS buffer (PBS 1× [#D8537; Sigma-Aldrich], bovine serum albumin 1% [#A7030-100G; Sigma-Aldrich], and EDTA 1 mM [#A1104, 0500; AppliChem GmbH]). Tubes were centrifuged at 4°C for 5 min at 300$g$. Supernatants were removed carefully and pellet was resuspended in 2 ml of a red blood cells lysis buffer (NH$_4$Cl 0.828%, KHCO$_3$ 0.1%, and EDTA 77.5 $\mu$M, filtered on a 0.22-$\mu$m filter) for 2 min. The cells were washed in FACS buffer as previously indicated and then the supernatant was resuspended again in 1 ml of lysis buffer for 1 min. After a final washing step, the cells were resuspended in 400 $\mu$l of FACS buffer and 200 $\mu$l was used for the assay.

Spleens were manually dissociated on 0.7-$\mu$m cell strainers (#352350; Corning) and washed in FACS buffer, as previously indicated. Pellets were resuspended in 1 ml red blood cell lysis buffer for 1 min and then washed again in FACS buffer. Pellets were resuspended in 10 ml FACS buffer. 100 $\mu$l of cell suspension was used for the assay.

Cervical and mesenteric lymph nodes were mechanically dissociated on 0.7 $\mu$m cell strainers and washed in FACS buffer, as previously indicated. Pellets were re-suspended in 400 $\mu$l of FACS buffer. 100 $\mu$l of cell suspension was used for the assay.

Ears were cut in very small pieces with a scalpel in a petri box cap and transferred in 3 ml of RPMI (#61870-010; Gibco, Thermo Fisher Scientific) FBS 10%, collagenase IV (#LS004188; Bioconcept) 1 mg/ml, DNAse (#79254; QIAGEN) 1/100. The suspension was incubated for 30 min at 37°C in a shaking water bath. The cells were put on a 0.7-$\mu$m filter, mechanically dissociated, and washed in FACS buffer as previously indicated before being resuspended in 300 $\mu$l of FACS buffer. 200 $\mu$l of this suspension was used for the assay.

Cells isolated for assay were plated in 96-well V-bottom plates (#05-021-0100; Boettger), washed in PBS 1×, and resuspended in 100 $\mu$l of Fixable Viability Life Stain 620 (#564996, 1/1,000; BD Biosciences) diluted in PBS. Cells were incubated for 15 min at room temperature and then washed twice in FACS buffer. Then, the cells were labeled with anti-CD16/CD32–purified antibody for 15 min at 4°C. Finally, the cells were labeled in 100 $\mu$l of antibody mixture diluted 1/2 in Brilliant Stain Buffer and 1/2 in FACS buffer, according to the manufacturer's instructions. The cells were incubated for 30 min at 4°C, washed twice in FACS buffer, transferred into 5-ml FACS tubes (#352058; Corning), and acquired on an LSR Fortessa (BD Biosciences).

Cell numbers in the prepared organs were determined with the Flow-Count Fluorospheres (#7547053; Beckman Coulter), following the manufacturer's instructions. Briefly, 50 $\mu$l of unlabeled cells from the initial cell suspensions were mixed with 50 $\mu$l of beads (bead concentration depended on the lot) and acquired on an LSR Fortessa (BD Biosciences). Initial cell counts were calculated with the following formula: (cell number counted/beads number counted) × beads concentration. Absolute numbers were

determined by normalizing the cell number counted after staining with antibodies to this calculated organ number.

Analyses were performed using the FlowJo software, version 10 (FlowJo LLC).

## Histopathological evaluation

Ears were fixed in 4% buffered formaldehyde and embedded in paraffin. Ear sections (4 $\mu$m) were deparaffinized and stained with hematoxylin and eosin (HE; Diapath S.p.A.). Slides were scanned on a Mirax Midi slide scanner (Carl Zeiss Microscopy). The ZEN blue software (Carl Zeiss Microscopy) was used for image acquisition and measurements. The average epidermal thickness was estimated by taking 20 measures along the ear.

## Keratinocyte isolation

For the isolation of primary keratinocytes, naïve mice were euthanized by carbon dioxide (CO$_2$) inhalation in euthanasia chamber. Primary murine skin keratinocytes were isolated from the tail of adult mice. Briefly, the tail was cut into four pieces and two or three incisions were made alongside the tail. Tails were rinsed in cold PBS (#D8537; Sigma-Aldrich)/100 U/ml penicillin/ 100 $\mu$g/ml streptomycin (P/S, #15070-063; Gibco, Thermo Fisher Scientific). The tail pieces were then incubated in Keratinocyte Serum-Free Medium (K-SFM) (#10725-018; Gibco, Thermo Fisher Scientific)/10 mg/ml Dispase II (#D4693; Sigma-Aldrich) overnight at 4°C. The epidermal layer was peeled away from the dermis with sterile forceps and the strips were gently mixed in trypsin/EDTA (P10-023100; Brunschwig). The cell suspension was collected in DMEM (#41966-029; Gibco, Thermo Fisher Scientific)—10% FBS—P/ S followed by filtration through a 70-$\mu$m cell strainer. After centrifugation (8 min at 200$g$), the cell pellet was resuspended in K-SFM complemented with 53.4 $\mu$g/ml bovine pituitary extract and 6.6 ng/ml human recombinant EGF (#37010-022; Gibco, Thermo Fisher Scientific)/P/S. The cells (2.5–5 × 10$^5$ cells/well) were cultured in collagen type IV (#C5533; Sigma-Aldrich) (200 $\mu$g/ml) coated 48-well tissue culture plate (#353078; Falcon, Thermo Fisher Scientific) at 37°C/5% CO$_2$. The following day, the medium was refreshed. On day 3, primary murine keratinocytes were stimulated with 100 ng/ml murine IL-36$\beta$ (generously given by Amgen Inc.) or 100 ng/ml murine IL-1$\beta$ (#211-11B; Peprotech) in K-SFM medium/P/S for 6 h, and untreated cells were used as a control.

## BMDCs

Femur and tibia of euthanized mice were collected and flushed with a syringe full of cold PBS 1× (#D8537; Sigma-Aldrich). Collected cells were filtered through a 0.7-$\mu$m cell strainer (#352350; Corning), centrifuged at 4°C for 5 min at 300$g$, and seeded into a P10 Petri dish (#353003; Corning) containing 10 ml of BMDC medium (RPMI [#61870-010; Gibco, Thermo Fisher Scientific], 10% FBS [#P40-37500; PAN-Biotech], 2-mercaptoethanol 50 $\mu$M [#161-0710; Bio-Rad], penicillin 100 U/ml, and streptomycin 100 $\mu$g/ml [P/S, #15070-063; Gibco, Thermo Fisher Scientific]). The cells were incubated at 37°C for 30 min. Supernatant was collected and centrifuged. The

pellet was resuspended in 10 ml of BMDC medium supplemented with 20 ng/ml of recombinant murine GM-CSF (#12343122; ImmunoTools) and transferred in a P10 Petri dish. The cells were cultured at 37°C, 5% $CO_2$, in a humidified incubator ($CO_2$ incubator model 371; Steri-Cycle, Thermo Fisher Scientific).

After 3 d, 5 ml of supernatant was collected and centrifuged. The pellet was resuspended in 10 ml of BMDC medium with GM-CSF and seeded into a new P10 Petri dish. The same operation was repeated at the sixth day of culture.

At the seventh day of culture, floating cells were removed. Adherent cells were washed once with PBS 1× (#D8537; Sigma-Aldrich) and then scraped, collected, washed, and resuspended in BMDC medium without GM-CSF. The purity of the BMDCs was determined by staining with anti-CD11b and anti-CD11c antibodies (see Antibodies section), following the FACS staining protocol as described above. Subsequently, the cells were stimulated with LPS (10 ng/ml [#L2880; Sigma-Aldrich]), murine-recombinant IL-36$\beta$ (100 ng/ml), or BMDC medium without GM-CSF, alone, for 24 h.

## RNA extraction and RT-quantitative (q)PCR

Total RNA was extracted from ears, spleen, lung, heart, liver, and cultured keratinocytes or BMDCs using TRIzol reagent (#15596018; Life Technologies, Thermo Fisher Scientific) and chloroform (#25669-1L; Sigma-Aldrich) for gradient phase separation. RNA was then purified using the RNeasy Mini Kit (#74104; QIAGEN), with treatment with RNase-free DNase (#79254; QIAGEN) according to the manufacturer's instructions. Total RNA (100–500 ng) was then reverse-transcribed using the SuperScript II Reverse Transcriptase (#18064-014; Life Technologies, Thermo Fisher Scientific). mRNA expression levels were determined by qRT-PCR using the SYBR Green PCR Master Mix (#A25776; Applied Biosystems, Thermo Fisher Scientific) according to the manufacturer's protocol. Sequences of the primers are as follows: Il1rl2: forward: AAACACCTAGCAAAAGCCCAG, reverse: AGACTGCCCGATTTTCCTATG. Intra-cytoplasmic Il1rn: forward: AGTGAGACGTTGGAAGGCAG, reverse: TGAAGGCTTGCATCTTGCAG. Secreted (s) Il1rn: forward: TGTGGCCTCGGGATGGAAAT, reverse: ATGAGCTGGTTGTTTCTCAGGT. Cxcl1: forward: ACTCAAGAATGGTCGCGAGG, reverse: GTGCCATCAGAGCAGTCTGT. Il23a: forward: CCAGCGGGACATATGAATCTACT, reverse: CTTGTGGGTCACAACCATCTTC. S100a9: forward: CACCCTGAGCAAGAAGGAAT, reverse: TGTCATTTATGAGGGCTTCATTT. Csf2: forward: CTCACCCATCACTGTCACCC, reverse: TGAAATTGCCCCGTAGACCC. Nfkbiz variant 3: forward: TGAAGTCCCGAGGTTGAGCC, reverse: TTCGGATGTGTAGCAGGAGC. Cxcl2: forward: AGGGCGGTCAAAAAGTTTGC, reverse: CGAGGCACATCAGGTACGAT. Csf3: forward: TGTCCTGGCCATTTCGTACC, reverse: AAGGGCTTTCTGCTCAGGTC. Il17a: forward: ATCCCTCAAAGCTCAGCGTGTC, reverse: GGGTCTTCATTGCGGTGGAGAG. Il22: forward: TTGACACTTGTGCGATCTCTGA, reverse: CGGTTGACGATGTATGGCTG. Relative levels of mRNA expression were normalized to L32 (Rpl32) mRNA levels and analyzed using the $2^{\Delta Ct}$ technique. Non-reverse-transcribed RNA samples and buffer were used as negative controls.

## Measurement of cytokine levels

Keratinocyte culture supernatants (see above) were collected, and CXCL1 concentrations were assessed with the DuoSet ELISA for CXCL1/KC kit (#DY453-05; R&D systems, Bio-Techne AG), following the manufacturer's instructions.

BMDC culture supernatants (see above) were collected and IL-6 concentrations assessed with the mIL-6 ELISA kit (#88–706; eBioscience, Thermo Fisher Scientific) following the manufacturer's instructions.

## RNAseq

mRNA was prepared from ears of Aldara-treated mice with the mirVana kit (#AM1560; Invitrogen, Thermo Fisher Scientific) following the manufacturer's instructions.

RNA cleanup was performed with RNA Clean XP beads (#A66514; Beckman Coulter). RNA quantification was performed with a Qubit fluorimeter (Life Technologies, Thermo Fisher Scientific) and RNA integrity assessed with a Bioanalyzer (Agilent Technologies). The TruSeq mRNA stranded kit from Illumina was used for the library preparation with 500 ng of total RNA as input. Library molarity and quality was assessed with the Qubit and TapeStation using DNA High Sensitivity Chip (Agilent Technologies). The 34 and 40 libraries were pooled at 2 nM and loaded for clustering on four and five lanes, respectively, of a Single-read Illumina Flow cell. Reads of 50 bases were generated using the TruSeq SBS chemistry on an Illumina HiSeq 4000 sequencer.

A unique gene model was used to quantify reads per gene. Briefly, the model considers all annotated exons of all annotated protein-coding isoforms of a gene to create a unique gene where the genomic region of all exons is considered coming from the same RNA molecule and merged together.

All reads overlapping the exons of each unique gene model were reported using featureCounts version 1.4.6-p1 (Quinlan & Hall, 2010). Gene expressions were reported as raw counts and in parallel normalized in RPKM to filter out genes with low expression value (1 RPKM) before calling for differentially expressed (DE) genes. Library size normalizations and differential gene expression calculations were performed using the package edgeR (Robinson et al, 2010) designed for the R software R Development Core Team, 2011. Only genes having a significant fold-change (Benjamini–Hochberg corrected $P$-value < 0.05) were considered for the rest of the RNAseq analysis.

We first defined IL-36–dependent genes, using the strategy depicted in Fig S5. Briefly, we selected the genes that were DE in Aldara-treated (AL) compared with untreated (CT) ears of WT mice. We obtained 3,088 genes (1,473 up-regulated and 1,615 down-regulated) at d3 and 4,795 genes (2,125 up-regulated and 2,670 down-regulated) at d7. Among these genes, we selected those that were non-DE (nDE) in AL compared with CT ears of KO mice, but DE in AL ears from WT compared with KO mice. These genes were termed IL-36 dependent.

Then, we defined the genes that were specifically regulated by IL-36R signaling in keratinocytes (termed KC36 dependent). For this purpose, we used the strategy depicted in Fig S6. We focused on the genes that were found DE in AL compared with CT ears of FL mice. We identified 3,168 genes (1,530 up-regulated and 1,638 down-regulated) at d3 and 4,689 genes (2,207 up-regulated and 2,482 down-regulated) at d7. Among these genes, we selected those that were IL-36 dependent as defined above. We obtained 200 and 600 genes at d3 and d7, respectively. Among these genes, we isolated

those that were nDE in AL and CT ears from DK mice, and DE in AL ears from FL and DK mice.

Finally, we selected the top 30 up- and down-regulated genes of each category at d3 and d7. We compared RPKM raw data to confirm that they were DE. A P-value < 0.05 as determined by Mann–Whitney test was considered significant. Gene expression values that did not show statistically significant difference when comparing AL and CT ears from both WT and FL mice were excluded. Then, values that showed statistically significant difference in AL ears from WT and KO mice were confirmed as IL-36 dependent. Finally, genes that showed statistically significant difference in AL ears from FL and DK mice were defined as KC36 dependent.

## Statistical analysis

Data are represented in mean value ± SEM. Statistical analysis was performed with the Prism software, versions 6–8 (GraphPad Software). Different statistical tests were used depending on the experimental conditions (paired or unpaired, normal distribution or not, two or more than two categories). Briefly, we used the following tests: two-way ANOVA followed by Holm–Sidak's comparison or mixed-effects analysis, unpaired $t$ test with Welch's correction, Mann–Whitney test, Wilcoxon matched pairs sign-ranked test, one-way ANOVA, or Friedman test with multiple comparisons. Details on the specific test used are given in the figure legends. A P-value < 0.05 was considered as significant.

# Data Availability

The RNASeq data from this publication have been deposited to the Gene Expression Omnibus database (https://www.ncbi.nlm.nih.gov/geo/query/acc.cgi?acc=GSE143688) and assigned the identifier GSE143688.

# Supplementary Information

# Acknowledgements

This work was supported by the grant from the Swiss National Science Foundation No 310030_712674 and a grant from the Rheumasearch Foundation. K5Cre mice were kindly provided by Dr Vincent Flacher (Institute of Cellular and Molecular Biology, Strasbourg, France). The authors would like to thank Teddy Verstraete, Paolo Quirighetti, and Stephanie Lafay from the animal facility of the Faculty of Medicine; Mylène Docquier and Didier Chollet from the genomic platform of the Faculty of Medicine; Cécile Gameiro from the flow cytometry platform; Christelle Veyrat-Durebex from the platform of animal surgery of the Faculty of Medicine; our actual and former lab members Gaby Palmer, Mathilde Harel, Loïc Mermoud, Praxedis Martin, Dominique Talabot, Alejandro Diaz Barrero, Sabina Troccaz, and Charlotte Girard; and our colleagues from the Department of Pathology and Immunology Carlo Chizzolini, Niccolo Brembilla, Luisa Senra, Marie Ballester, and Stéphanie Hugues for useful discussions and help with protocols.

## Authors Contributions

JD Goldstein: conceptualization, data curation, formal analysis, validation, investigation, methodology, and writing—original draft.
EY Bassoy: data curation, formal analysis, investigation, and writing—original draft.
A Caruso: data curation and investigation.
J Palomo: conceptualization, data curation, formal analysis, and investigation.
E Rodriguez: data curation and investigation.
S Lemeille: data curation, software, formal analysis, and visualization.
C Gabay: conceptualization, resources, data curation, supervision, funding acquisition, validation, visualization, methodology, project administration, and writing—review and editing.

## Conflict of Interest Statement

The authors declare that they have no conflict of interest.

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
