## [Reviewer comments · Life Science Alliance]

Life Science Alliance

IL-36 signaling in keratinocytes controls early IL-23 production in psoriasis-like dermatitis

Jeremie Goldstein, Esen Bassoy, Assunta Caruso, Jennifer Palomo, Emiliana Rodriguez, Sylvain Lemeille, and Cem Gabay

DOI: <https://doi.org/10.26508/lsa.202000688>

Corresponding author(s): Cem Gabay, University of Geneva, Faculty of Medicine

Review Timeline:

Submission Date:	2020-02-28
Editorial Decision:	2020-04-03
Revision Received:	2020-04-10
Editorial Decision:	2020-04-14
Revision Received:	2020-04-15
Accepted:	2020-04-16

Scientific Editor: Andrea Leibfried

Transaction Report:

April 3, 2020

Re: Life Science Alliance manuscript #LSA-2020-00688

Prof. Cem Gabay
University of Geneva, Faculty of Medicine
Pathology and Immunology
Centre Médical Universitaire
Rue Michel-Servet 1
Geneva 1206
Switzerland

Dear Dr. Gabay,

Thank you for submitting your manuscript entitled "IL-36 signaling in keratinocytes controls early IL-23 production in Aldara®-induced murine psoriasis" to Life Science Alliance. The manuscript was assessed by expert reviewers, whose comments are appended to this letter.

As you will see, while reviewer #1 thinks that your work does not add much to the existing literature, reviewer #2 and #3 are more positive on your work and provide constructive input on how to further strengthen your manuscript. Most concerns raised can get addressed by responding to them and changing the manuscript text accordingly. Two experiments (Western blot and ELISA) are requested. Given the current pandemic, I would be happy to discuss with you how you could address these specific requests made. Maybe we can speak on the phone or via skype, or you can send me a point-by-point response outline via response email.

Thank you for this interesting contribution to Life Science Alliance. We are looking forward to receiving your revised manuscript.

Sincerely,

B. MANUSCRIPT ORGANIZATION AND FORMATTING:

Reviewer #1 (Comments to the Authors (Required)):

The authors investigated role(s) of IL-36 receptor signaling in the pathogenesis of Aldara-induced psoriasis-like skin lesions. They revealed that IL-36 receptor signaling in keratinocytes induced neutrophil infiltration in Aldara-treated ears and IL-23, IL-17 and IL-22 expression. The authors

concluded that IL-36 receptor signaling in keratinocytes plays a crucial role in the pathogenesis of Aldara-induced skin inflammation.

I do not understand the reason why the authors studied Aldara-induced skin inflammation. Are the Aldara-induced skin lesions an important problem in dermatology or skin biology? There are many studies on IL-36 receptor signaling in psoriasis in the literature. Imiquimod-induced psoriasis models have been used in a number of the studies. Do the authors think Aldara cream has specific effects to the skin? Do contents other than imiquimod play a role in the pathogenesis of Aldara-induced skin lesions?

Reviewer #2 (Comments to the Authors (Required)):

This is a well written manuscript reporting the contribution of the interleukin-36 receptor pathway in keratinocytes, in the imiquimod-induced psoriasis-like model. Previous studies of the same model led to conclusions that the IL36 pathway was crucial for skin inflammation induced by imiquimod, a Toll-like Receptor 7 (TLR7) agonist. The added value of this study is to dissect specifically the role of the keratinocyte compartment in the IL36-driven cascade, using robust invalidation methods making sure that the IL36R signaling is completely invalidated at the keratinocytic level, while other groups used anti-IL36 receptor antibodies, a less robust methodology which does not allow to specifically address the contribution of keratinocytes. This manuscript appeals several comments :

- The imiquimod induced psoriasiform dermatitis shares some features with chronic psoriasis vulgaris at the cellular and cytokine level, especially the upregulation of the IL23-IL17, TNF alpha and IL22 cytokines, but diverges by intense upregulation of type I interferon pathway and interferon-stimulated gene (which is only detected in human psoriasis at the acute, plaque formation phase), and by a more spectacular recruitment of neutrophils. In line with these specific features, I suggest to rephrase the title and to name this skin model « psoriasis-like dermatitis ». Also, as other works have shown that IL36 agonists were able to enhance the transcription of interferon type I-related genes, it would be interesting to have a specific address and comments regarding the consequences of IL36R invalidation on type I interferon pathway signalling at the keratinocyte level, although the major source of type IFN remain plasmacytoid dendritic cells.
- In the introduction section, the authors claim that IL36 agonist cytokines are produced, although to a lesser extent, by other cells than keratinocytes including macrophages and neutrophils. Actually, it is not quite clear if neutrophils truly express the transcripts and autonomously release the bioactive IL36 cytokines, but there is compelling evidence that myeloid cells are key in the IL36 pathway activation through their production of proteases, cathepsin G, elastase and proteinase 3, which are key in cleaving the immature forms of IL36 agonists (and antagonists for elastase) into bioactive forms. Then comes a natural query regarding the authors' study : if the recruitment of neutrophils is so much inhibited in keratinocyte-specific invalidated IL36R, it would be of major added value to document (through western blot), the absence of bioactive IL36 alpha at the protein level, which is expected.
- The remaining acanthosis in IL36R-keratinocyte-specific invalidated animals is very compelling, and of special interest regarding the mechanistic scenario in human psoriasis and pustular psoriasis, as some data are suggestive for a role of IL36 in keratinocyte proliferation. Did the authors perform proliferation markers immunostaining analysis (ki67) in these animals to check for the presence or absence of hyperproliferation markers ? In the absence of Ki67 increased expression in keratinocyte-specific IL36R invalidated animals, it would clearly raise another scenario for keratinocyte homeostasis deregulation in IL36-driven skin inflammation, not only in psoriasis.

- The number of pDCs (D3 and D7) and monocytes (D3) are not different between FL and DK mice, but did the authors check for expression of type I interferon by pDC in respective models? Again, this might challenge (or not) the contribution of type I interferon in the imiquimod mouse model. Regarding monocytes, it would be interesting to show the actual production of both IL36 agonists (bioactive forms eventually) and of cathepsin G in DK mice.
- The discussion is thoughtful and balanced, however I suggest to include in it references to key cytokines in acute psoriatic inflammation (and IL23 at early time point might not be the major driver), such as type I interferon.

Reviewer #3 (Comments to the Authors (Required)):

The authors present a very interesting and exciting manuscript. The group are a world leading authority on this topic. This is mirrored by the fact that the experiments conducted, the use of correct controls and interpretation of the data has been done to a very high standard. The authors should also be highly commended for their professionalism of highlight a "rival" manuscript in the same journal that has just been published.

My corrections are only minor.

1) Methods and results. Many readers of the manuscript will have little or lay knowledge of genetics and mice crossing methods. Especially the rationale to use r K5Cre or K14Cre mice. This should be explained better, either in the methods or in the sup material.

2) The finding that IL-36 may induce IL-23 from keratinocytes is of great interest and could be central to the early disease mechanism. A Previous report (also correctly references in this paper) also reported this. However, a common problem with measuring IL-23, is that IL-23 measurement by gene expression is no longer reliable. Gene expression only measures IL-23p19 and not full IL-23 protein. A new cytokine, that has been named IL-39 consists of EBI3 and p19. While this is controversial, this chain pairing has been confirmed in human keratinocytes, and is upregulated following inflammatory stimuli. See references:

Tachibana, K., et al. "371 The expression of p19 and EBI3 in epidermal keratinocytes under the stimulation with inflammatory cytokines." *Journal of Investigative Dermatology* 139.9 (2019): S278.
Ramnath, Divya, et al. "TLR3 drives IRF6-dependent IL-23p19 expression and p19/EBI3 heterodimer formation in keratinocytes." *Immunology and cell biology* 93.9 (2015): 771-779.

Thus upregulation of IL-23p19, does not equate to IL-23, and this may be especially so in keratinocytes. It would be strongly advisable that the authors, culture keratinocytes, stimulate with IL-36 and measure the full IL-23 protein using ELISA.

3)The authors should make reference in the discussion to another paper.

Costa, Sara, et al. "Role of MyD88 signaling in the imiquimod-induced mouse model of psoriasis: focus on innate myeloid cells." *Journal of leukocyte biology* 102.3 (2017): 791-803.

In this paper myd88 has selective KO on monocytes, and this stops severity of IMQ inflammation, but it does not stop initiation and epidermal changes. This myd88 signalling may well be IL-36 driven, so should be included in discussion, as it potentially compliments the authors findings.

4) Dermal gamma delta cells are seen at the main IL-17+ T cells in IMQ model. The findings of the authors differ, it would be of great interest to confirm the source of IL-17 in their paper. CD4, CD8,

Neutrophils, ILC3?

Reviewer #1 (Comments to the Authors (Required)):

The authors investigated role(s) of IL-36 receptor signaling in the pathogenesis of Aldara-induced psoriasis-like skin lesions. They revealed that IL-36 receptor signaling in keratinocytes induced neutrophil infiltration in Aldara-treated ears and IL-23, IL-17 and IL-22 expression. The authors concluded that IL-36 receptor signaling in keratinocytes plays a crucial role in the pathogenesis of Aldara-induced skin inflammation.

I do not understand the reason why the authors studied Aldara-induced skin inflammation. Are the Aldara-induced skin lesions an important problem in dermatology or skin biology? There are many studies on IL-36 receptor signaling in psoriasis in the literature. Imiquimod-induced psoriasis models have been used in a number of the studies. Do the authors think Aldara cream has specific effects to the skin? Do contents other than imiquimod play a role in the pathogenesis of Aldara-induced skin lesions?

Aldara cream contains 5% imiquimod and has been used as a model of psoriasis in several studies (Cai Y et al. 2011. Immunity ; Tortola L et al. 2012. J Clin Invest), including the initial study defining this mouse model of psoriasis (Van der Fits et al. 2009. J Immunol). We have been using this cream in some of our own previous studies (Palomo J et al. 2018. PLoS One ; Martin P et al. 2020. J Immunol). The objective of our study was to determine the role of IL-36R in keratinocytes in comparison with full IL-36R deficiency as previously demonstrated (Tortola L et al. 2012. J Clin Invest). In addition to Imiquimod, the Aldara cream contains isostearic acid, which exerts some of the proinflammatory effects of Aldara effects on skin pathogenesis (Walter A et al. 2013. Nat Commun; Martin P et al. unpublished observations). Thus, we used the term "Aldara-induced psoriasis" rather than "imiquimod-induced psoriasis" throughout our manuscript.

Reviewer #2 (Comments to the Authors (Required)):

This is a well written manuscript reporting the contribution of the interleukin-36 receptor pathway in keratinocytes, in the imiquimod-induced psoriasis-like model. Previous studies of the same model led to conclusions that the IL36 pathway was crucial for skin inflammation induced by imiquimod, a Toll-like Receptor 7 (TLR7) agonist. The added value of this study is to dissect specifically the role of the keratinocyte compartment in the IL36-driven cascade, using robust invalidation methods making sure that the IL36R signaling is completely invalidated at the keratinocytic level, while other groups used anti-IL36 receptor antibodies, a less robust methodology which does not allow to specifically address the contribution of keratinocytes. This manuscript appeals several comments :

- The imiquimod induced psoriasiform dermatitis shares some features with chronic psoriasis vulgaris at the cellular and cytokine level, especially the upregulation of the IL23-IL17, TNF alpha and IL22 cytokines, but diverges by intense upregulation of type I interferon pathway and interferon-stimulated gene (which is only detected in human psoriasis at the acute, plaque formation phase), and by a more spectacular recruitment of neutrophils. In line with these specific features, I suggest to rephrase the title and to name this skin model « psoriasis-like dermatitis ».

Thank you for the suggestion.

Also, as other works have shown that IL36 agonists were able to enhance the transcription of interferon type I-related genes, it would be interesting to have a specific address and comments regarding the consequences of IL36R invalidation on type I interferon pathway signalling at the keratinocyte level, although the major source of type IFN remain plasmacytoid dendritic cells.

We will modify the Results and Discussion sections to include the results of RNASeq analysis on regulation of genes from type I interferon signaling pathway.

- In the introduction section, the authors claim that IL36 agonist cytokines are produced, although to a lesser extent, by other cells than keratinocytes including macrophages and neutrophils. Actually, it is not quite clear if neutrophils truly express the transcripts and autonomously release the bioactive IL36 cytokines, but there is compelling evidence that myeloid cells are key in the IL36 pathway activation through their production of proteases, cathepsin G, elastase and proteinase 3, which are key in cleaving the immature forms of IL36 agonists (and antagonists for elastase) into bioactive forms. Then comes a natural query regarding the authors' study : if the recruitment of neutrophils is so much inhibited in keratinocyte-specific invalidated IL36R, it would be of major added value to document (through western blot), the absence of bioactive IL36 alpha at the protein level, which is expected.

The processing of IL-36 α into its active form is performed by neutrophil-derived cathepsin G and elastase (Henry CM et al. 2016. Sci Rep). Cathepsin G cleaves IL-36 α at the residue Lys 3, and elastase at the residue Ala4. Thus, the bioactive form of IL-36 α is too similar in size with its inactive counterpart for allowing clear distinction by Western blot analysis. Thus, we did not perform the experiment suggested by the reviewer. Nevertheless it is plausible that, as suggested by the reviewer, IL-36alpha maturation is decreased in DK mice, thus further decreasing the stimulatory effects of IL-36 on other target cells in the skin than in keratinocytes. We modified the Discussion section to address this point.

- The remaining acanthosis in IL36R-keratinocyte-specific invalidated animal is very compelling, and of special interest regarding the mechanistic scenario in human psoriasis and pustular psoriasis, as some data are suggestive for a role of IL36 in keratinocyte proliferation. Did the authors perform proliferation markers immunostaining analysis (ki67) in these animals to check for the presence or absence of hyperproliferation markers ?

In a previous study we did perform Ki67 staining in WT mice and showed that the staining was present predominantly in the basal layer during Aldara-induced psoriasis. However, we did not perform this analysis in this study.

In the absence of Ki67 increased expression in keratinocyte-specific IL36R invalidated animals, it would clearly raise another scenario for keratinocyte homeostasis deregulation in IL36-driven skin inflammation, not only in psoriasis.

- The number of pDCs (D3 and D7) and monocytes (D3) are not different between FL and DK mice, but did the authors check for expression of type I interferon by pDC in respective models ?

No, we did not perform any cell-specific cytokine expression or production assay apart from keratinocytes and bone marrow-derived dendritic cells. However, RNASeq experiments did not indicate any significant increase of expression of *Ifna* or *Ifnb* in Aldara-treated or untreated skin both at days 3 or 7. This result is consistent with a previous study, in which the authors did not detect any increased production of IFN- α protein in the skin over a treatment course of four days, despite increased serum IFN- α and IFN- β concentrations 3 hours after the first Aldara application (Grine L et al. 2015. J Immunol). As indicated by Reviewer #2, this finding could be related to the difference between the human disease and Aldara-treated mice, with patients presenting a much higher production of type-I interferons and associated proteins in the early events of psoriasis than Aldara-treated mice.

Again, this might challenge (or not) the contribution of type I interferon in the imiquimod mouse model. Regarding monocytes, it would be interesting to show the actual production of both IL36 agonists (bioactive forms eventually) and of cathepsin G in DK mice.

Cathepsin G expression could not be detected by RNASeq at neither d3 nor d7, indicating that the level of cathepsin G expression might be too low for protein detection as well or that its regulation is not at the level of gene transcription. RNASeq also indicated that *Il1f6* (coding for IL-36 α), *Il1f8* (IL-36 β) and *Il1f9* (IL-36 γ) were induced by Aldara treatment in all mice strains at both d3 and d7. *Il1f6* and *Il1f9* expression were partly dependent of IL-36 signaling at d3 and d7, whereas *Il1f8* expression was only partly dependent of IL-36 signaling at d7. None of IL-36 agonists were regulated by an IL-36 signaling in keratinocytes at neither time points. Concerning the detection of bioactive forms, the same remark than previously applies, all IL-36 cytokines having in common the property of being cleaved at their N-terminal extremity (Bassoy E et al. 2018. Immunol Rev) leading to minor changes in size despite major changes in bioactivity. Thus, detection by Western blot of bioactive forms might prove technically challenging.

- The discussion is thoughtful and balanced, however I suggest to include in it references to key cytokines in acute psoriatic inflammation (and IL23 at early time point might not be the major driver), such as type I interferon.

Thank you for the suggestion. The discussion will be updated accordingly.

Reviewer #3 (Comments to the Authors (Required)):

The authors present a very interesting and exciting manuscript. The group are a world leading authority on this topic. This is mirrored by the fact that the experiments conducted, the use of correct controls and interpretation of the data has been done to a very high standard. The authors should also be highly commended for their professionalism of

highlight a "rival" manuscript in the same journal that has just been published.

My corrections are only minor.

1) Methods and results. Many readers of the manuscript will have little or lay knowledge of genetics and mice crossing methods. Especially the rationale to use r K5Cre or K14Cre mice. This should be explained better, either in the methods or in the sup material.

We thank the reviewer for pointing this out. Materials and Methods section will be revised accordingly.

2) The finding that IL-36 may induce IL-23 from keratinocytes is of great interest and could be central to the early disease mechanism. A Previous report (also correctly references in this paper) also reported this. However, a common problem with measuring IL-23, is that IL-23 measurement by gene expression is no longer reliable. Gene expression only measures IL-23p19 and not full IL-23 protein. A new cytokine, that has been named IL-39 consists of EBI3 and p19. While this is controversial, this chain pairing has been confirmed in human keratinocytes, and is upregulated following inflammatory stimuli. See references:

Tachibana, K., et al. "371 The expression of p19 and EBI3 in epidermal keratinocytes under the stimulation with inflammatory cytokines." *Journal of Investigative Dermatology* 139.9 (2019): S278.

Ramnath, Divya, et al. "TLR3 drives IRF6-dependent IL-23p19 expression and p19/EBI3 heterodimer formation in keratinocytes." *Immunology and cell biology* 93.9 (2015): 771-779.

Thus upregulation of IL-23p19, does not equate to IL-23, and this may be especially so in keratinocytes. It would be strongly advisable that the authors, culture keratinocytes, stimulate with IL-36 and measure the full IL-23 protein using ELISA.

Thank you for this very interesting point. We have checked RNASeq results for *Ebi3* expression. We observed that *Ebi3* was induced by Aldara at d7, and independently of IL-36 signaling, but *Ebi3* expression levels were really low. At d3, where we highlighted the role of IL-36 signaling for IL-23 expression, *Ebi3* could not be detected by RNASeq analysis, while expression of the second subunit of IL-23, *Il12b*, was among the top 30 genes up-regulated by IL-36 signaling in keratinocytes. These results suggest that it is indeed IL-23 but not IL-39, that is the active molecule upregulated by IL-36R signaling in keratinocytes at d3 of Aldara treatment. The Discussion was modified to address this important point.

3)The authors should make reference in the discussion to another paper.

Costa, Sara, et al. "Role of MyD88 signaling in the imiquimod-induced mouse model of psoriasis: focus on innate myeloid cells." *Journal of leukocyte biology* 102.3 (2017): 791-803.

In this paper myd88 has selective KO on monocytes, and this stops severity of IMQ inflammation, but it does not stop initiation and epidermal changes. This myd88 signalling may well be IL-36 driven, so should be included in discussion, as it potentially compliments the authors findings.

We thank the reviewer for his suggestion and will modify the Discussion accordingly.

4) Dermal gamma delta cells are seen at the main IL-17+ T cells in IMQ model. The findings of the authors differ, it would be of great interest to confirm the source of IL-17 in their paper. CD4, CD8, Neutrophils, ILC3?

We agree with the reviewer that it would be important to identify the cell sources of IL-17. However, this interesting point is beyond the scope of the current manuscript.

April 14, 2020

RE: Life Science Alliance Manuscript #LSA-2020-00688R

Prof. Cem Gabay
University of Geneva, Faculty of Medicine
Pathology and Immunology
Centre Médical Universitaire
Rue Michel-Servet 1
Geneva 1206
Switzerland

Dear Dr. Gabay,

Thank you for submitting your revised manuscript entitled "IL-36 signaling in keratinocytes controls early IL-23 production in psoriasis-like dermatitis". I have now assessed the revisions performed and think that they address the concerns of the reviewers in a good way. I would thus be happy to publish your paper in Life Science Alliance pending final revisions necessary to meet our formatting guidelines.

- Please link your ORCID iD to your profile in our submission system, you should have received an email with instructions on how to do so
- Please upload all figures, including supplementary figures, as individual files; when preparing the production-quality figures, please keep in mind that text should be easily readable at maximum figure size (7 in wide x 9 in high (17.5 x 22.8 cm))
- Please provide Table 1 and S2 as docx or excel files

A. FINAL FILES:

-- High-resolution figure, supplementary figure and video files uploaded as individual files: See our detailed guidelines for preparing your production-ready images, <http://www.life-science->

alliance.org/authors

B. MANUSCRIPT ORGANIZATION AND FORMATTING:

Sincerely,

Andrea Leibfried, PhD
Executive Editor
Life Science Alliance
Meyershofstr. 1
69117 Heidelberg, Germany
t +49 6221 8891 502
e a.leibfried@life-science-alliance.org

April 16, 2020

RE: Life Science Alliance Manuscript #LSA-2020-00688RR

Prof. Cem Gabay
University of Geneva, Faculty of Medicine
Pathology and Immunology
Centre Médical Universitaire
Rue Michel-Servet 1
Geneva 1206
Switzerland

Dear Dr. Gabay,

Thank you for submitting your Research Article entitled "IL-36 signaling in keratinocytes controls early IL-23 production in psoriasis-like dermatitis". It is a pleasure to let you know that your manuscript is now accepted for publication in Life Science Alliance. Congratulations on this interesting work.

*****IMPORTANT:** If you will be unreachable at any time, please provide us with the email address of an alternate author. Failure to respond to routine queries may lead to unavoidable delays in publication.*******

DISTRIBUTION OF MATERIALS:

Again, congratulations on a very nice paper. I hope you found the review process to be constructive and are pleased with how the manuscript was handled editorially. We look forward to future exciting

submissions from your lab.

Sincerely,
